# Resonance in Weight Space: Covariate Shift Can Drive Divergence of SGD with Momentum

**Kirby Banman[1], Liam Peet-Pare[2], Nidhi Hegde[1], Alona Fyshe[1], Martha White[1]**
Department of Computing Science[1], Department of Mathematical and Statistical Sciences[2]
University of Alberta
Edmonton, AB T6G 2E8
{kdbanman, peetpare, nidhi.hegde, alona, whitem}@ualberta.ca

## Abstract

Most convergence guarantees for stochastic gradient descent with momentum (SGDm) rely on iid sampling. Yet, SGDm is often used outside this regime, in settings with temporally correlated input samples such as continual learning and reinforcement learning. Existing work has shown that SGDm with a decaying step-size can converge under Markovian temporal correlation. In this work, we show that SGDm under covariate shift with a fixed step-size can be unstable and diverge. In particular, we show SGDm under covariate shift is a parametric oscillator, and so can suffer from a phenomenon known as resonance. We approximate the learning system as a time varying system of ordinary differential equations, and leverage existing theory to characterize the system's divergence/convergence as resonant/nonresonant modes. The theoretical result is limited to the linear setting with periodic covariate shift, so we empirically supplement this result to show that resonance phenomena persist even under non-periodic covariate shift, nonlinear dynamics with neural networks, and optimizers other than SGDm.

## 1 Introduction

Stochastic gradient descent (SGD) (Robbins & Monro, 1951) – and its variants such as Adagrad (Duchi et al., 2011), ADAM (Kingma & Ba, 2014) and RMSprop (Hinton et al., 2012) – are very widely used optimization algorithms across machine learning. SGD is conceptually straightforward, easy to implement, and often performs well in practice. Among the variants of SGD, accelerated versions based on Polyak's or Nesterov's acceleration (Polyak, 1964; Nesterov, 1983), known generally as Stochastic Gradient Descent with Momentum (SGDm), are used widely due to the improvements in convergence rate they offer. SGDm can give up to a quadratic speedup to SGD on many functions, and is in fact optimal among all methods having only information about the gradient at consecutive iterates for convex and Lipschitz continuous optimization problems (Nesterov, 2004; Goh, 2017). SGDm has the same computational complexity as SGD, but exhibits superior convergence rates under reasonable assumptions (Su et al., 2014).

These convergence results for SGDm, however, rely on independent and identically distributed (iid) sampling. Little is known about the convergence properties of SGDm under non-iid sampling, yet the non-iid setting is critical. In many machine learning problems it is expensive or impossible to obtain iid samples. In online learning (Rakhlin et al., 2010) and reinforcement learning (RL) (Sutton & Barto, 2018), the data becomes available in a sequential order and there is a temporal dependence among the samples. There is a particularly strong temporal dependence in RL, where observed states are sampled according to the transition dynamics of the underlying Markov decision process (MDP). Federated learning (Hsieh et al., 2020) and time-series learning (Kuznetsov & Mohri, 2014) provide further examples of when non-iid sampling is essential to the learning problem.

Without momentum, SGD's convergence rate has been examined under non-iid sampling with the stochastic approximation framework in (Benveniste et al., 2012; Kushner & Yin, 2003), and more recently under specific assumptions of ergodicity (Duchi et al., 2012) or Markovian sampling (Nagaraj et al., 2020; Doan et al., 2020b; Sun et al., 2018). Convergence rates under Markovian sampling are also known for ADAM-type algorithms when applied to policy gradient and temporal difference

learning (Xiong et al., 2020). To our knowledge, however, there has been little work on providing convergence rates or guarantees for SGDm under non-iid sampling. In (Doan et al., 2020a), a progress bound is provided for SGDm under Markovian sampling based on mixing time—the time required for a distribution's convergence toward its stationary distribution—along with a convergence rate guarantee under decaying step-sizes. In this work, we assume a fixed step-size, since it is a common choice in the online setting, especially when the practitioner is unsure of mixing rate or stationarity.

There is a broad literature using ordinary differential equations (ODEs) to analyze gradient descent methods by approximating descent updates as continuous-time flows. Early work is comprehensively discussed in (Kushner & Yin, 2003; Benveniste et al., 2012). Despite the age and establishment of the linear setting, the gradient flow lens continues to reveal new insights (e.g. implicit rank reduction (Arora et al., 2019)) and new perspectives on old insights (e.g. regularization of early stopping (Ali et al., 2019)). Recent work has paid particular attention to the flow induced by momentum accelerated methods (Su et al., 2014; Wibisono et al., 2016; Wilson et al., 2016; Scieur et al., 2017; Muehlebach & Jordan, 2021; Diakonikolas & Orecchia, 2019; Muehlebach & Jordan, 2019; Li et al., 2017; Simsekli et al., 2020; Betancourt et al., 2018; Kovachki & Stuart, 2019; Attouch et al., 2018; Shi et al., 2019; Siegel, 2019; Zhang et al., 2018; Berthier et al., 2021; Kovachki & Stuart, 2021). In these works, it is demonstrated that linear regression under iid sampling with SGDm can be represented as an ODE resembling a harmonic oscillator, sometimes with a time-decaying damping coefficient.

**Contributions:** In this work, we show that non-iid sampling, due to covariate shift, induces a related system: the parametric oscillator, a harmonic oscillator having coefficients which vary over time in a manner capable of exponentially exciting the system (Mumford, 1960; Csörgő & Hatvani, 2010; Halanay, 1966). We characterize the underlying time-varying ODE, which requires a novel approach to incorporate covariate shift, and develop specific conditions on covariate shift that lead to divergence in SGDm. We provide an empirical design, with synthetic data, to systematically test the response of the learning system to different covariate shift frequencies. We empirically validate that resonance-driven divergence occurs in a learning system which aligns well with theoretical assumptions. We follow with similar demonstrations on learning systems which progressively relax further and further away from our theoretical assumptions, including non-periodic covariate shift, learning with neural networks and optimizers other than SDGm. The implications of this work are that resonance can occur in learning systems with momentum, causing instability or poor accuracy, simply due to certain temporal correlations in the input sequence. Our results provide a novel direction to better understand the properties of our learning systems for time varying inputs—namely beyond iid sampling—and point to a gap in the robustness of our algorithms that merits further investigation.

## 2 PROBLEM SETTING

We investigate the effect of non-iid sampling on a model optimized using SGDm. We assume labelled training data sampled from discrete-indexed, real-valued stochastic processes $\{X_k\}_{k\in\mathbb{N}}$ and $\{Y_k\}_{k\in\mathbb{N}}$ such that $\{X_k\}_{k\in\mathbb{N}}$ has a unique stationary distribution $\Pi$. $Y_k$ is a function of $X_k$ with zero-mean observation noise, $Y_k = f(X_k) + \epsilon_k$. The learning algorithm does not have access to iid samples from $\Pi$. Instead, the learning algorithm may only update parameters $\theta_k$ at time $k$ using samples $z_k = (x_k, y_k)$, where $X_k$'s marginal over time converges to $\Pi$. The goal is to train a model with parameters $\theta$ to minimize an objective function $L(z;\theta)$ with respect to the stationary distribution $\Pi$: $\theta^* = \arg\min_{\theta\in\mathbb{R}^d}\mathbb{E}_\Pi[L(z;\theta)] = \int_\mathbb{R} L(z;\theta)d\Pi(x)$. An example of this problem setting is illustrated in Figure 1.

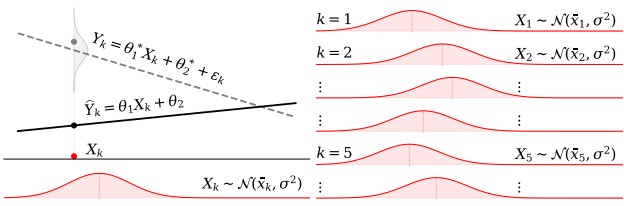

Figure 1: An example of the problem setting. 1D regression (left), Gaussian covariates $\{X_k\}$ with shifting mean $\bar{x}_k$ over time (right).

This setting is identical to that explored in prior work on Markovian sampling (Sun et al., 2018; Xiong et al., 2020; Nagaraj et al., 2020), but we do not require that our stochastic processes adhere to the Markov property. Note, also, that the underlying functional relationship between the inputs, $X_k$,

and the targets, $Y_k$, does not change over time. That is, the non-iid sampling is a result of covariate shift rather than a changing relationship between the inputs and targets.

We consider Polyak's Heavy Ball method (Polyak, 1964) with the formulation from Sutskever et al. (2013), namely SGD with momentum (SGDm)

$$v_{k+1} = \mu v_k - \eta \nabla_\theta L(z_k; \theta_k) \qquad\qquad \theta_{k+1} = \theta_k + v_{k+1} \qquad (1)$$

where $\eta \geq 0$ is the learning rate and $\mu \in (0, 1)$ the momentum coefficient. Our goal is to understand the behavior of SGDm when learning on this non-iid sequence $z_k$.

## 3    COVARIATE SHIFT AS A DRIVING FORCE

In this section we characterize SGDm as a discretization of a particular parametric oscillator in continuous time. A parametric oscillator resembles a harmonic oscillator ODE, but has time-varying system coefficients which are capable of driving the system. It is well understood that parametric oscillators can suffer from global solution instability due to coefficients oscillating at particular frequencies (Halanay, 1966), a condition known as *parametric resonance*. We show parametric resonance conditions sufficient to induce exponential divergence in SGDm. We provide proof sketches and defer complete proofs to Appendix A.4.

**Notation:** Capital letters are matrices and random variables. $\{X_k\}$ is a stochastic process, shorthand for $\{X_k\}_{k \in \mathbb{N}}$. $k$ and $t$ index discrete and continuous time, respectively. When a discrete sequence and a function approximate each other, they share the same symbol and are differentiated by subscript $k$ and function argument $t$, e.g. the sequence $\{\theta_k\}$ and the function $\theta(t)$. $P_k$ is the joint distribution of $X_k$ and $Y_k$. For loss function $L(z; \theta)$ on training pair $z = (x, y)$ and weights $\theta$, we denote the time-varying expected gradient $g_k(\theta) := \mathbb{E}_{P_k}[\nabla_\theta L(z; \theta)]$

### 3.1    COVARIATE SHIFT INDUCES A LINEAR TIME-VARYING EXPECTED GRADIENT

We begin by showing that linear least squares regression with covariate shift and a fixed target induces a linear time-varying expected loss gradient. That is, the gradient function $g_k(\theta) = B_k(\theta - \theta^*)$ for some matrix $B_k$ that varies with time, due to covariate shift.

**Assumption 1.** *The covariate generating process $\{X_k\}_{k \in \mathbb{N}}$ is not identically distributed: $Corr(X_{k_1}, X_{k_1}) \neq Corr(X_{k_2}, X_{k_2})$ for some $k_1, k_2 \in \mathbb{N}$.*

**Assumption 2.** *The targets $Y_k$ are a fixed linear function of $X_k$ with iid zero-mean observation noise, $\epsilon_k$ such that $\mathbb{E}[\epsilon_k] = 0$. That is, $Y_k = \langle \theta^*, X_k \rangle + \epsilon_k$ where $\theta^* \in \mathbb{R}^d$ is fixed for all $k$.*

**Proposition 1.** *Under Assumptions 1, 2, a linear model with weights $\theta_k$ making predictions $\widehat{Y}_k = \langle \theta_k, X_k \rangle$ with a mean squared error (MSE) objective will induce time-varying linear expected loss gradients $g_k(\theta_k) = B_k(\theta_k - \theta^*)$ for all $k \in \mathbb{N}$, where $B_k \in \mathbb{R}^{d \times d}$ and $B_{k_1} \neq B_{k_2}$ for some $k_1, k_2$.*

*Proof Sketch:* We fix a time step $k$ and take the expectation of the MSE loss over observation noise, and over the distribution of $X_k$. A time-varying expected loss surface remains, and the linear setting implies the loss gradient is a linear function for all steps $k$, with $B_k$ depending on $X_k$:

$$B_k = 2\text{Corr}(X_k, X_k) = 2\text{Cov}(X_k, X_k) + 2\mathbb{E}_{P_k}[X_k]\mathbb{E}_{P_k}[X_k]^T \qquad (2)$$

### 3.2    ODE CORRESPONDENCE

Next, we show that the iterates $\{\theta_k\}$ generated by SGDm are a first order numerical integration of a particular ODE. The procedure is similar to (Muehlebach & Jordan, 2021), but due to the time-varying loss gradient, we must pay specific attention to the conditions on the ODE necessary to have integration consistency, . Let $B(t)$ be a matrix-valued function, Lipschitz continuous in $t$, such that $\{B_k\}$ are samples spaced $\sqrt{\eta}$ apart, i.e. $B_k = B(\sqrt{\eta}k)$. The existence and uniqueness of this $B(t)$ are ensured if we assume the $\{X_k\}$ are sampled from an underlying continuous-time $X(t)$.

**Proposition 2.** *The SGDm iterates $\{\theta_k\}$ numerically integrate the ODE system in equation 3 with integration step $\sqrt{\eta}$ and first order consistency.*

$$\ddot{\theta}(t) + \frac{1 - \mu}{\sqrt{\eta}}\dot{\theta}(t) + B(t)(\theta(t) - \theta^*) = 0 \qquad (3)$$

*Proof Sketch:* First, the ODE in equation 3 is converted to first order linear form, and split into a sum of two separate systems. The sum's terms have implicit and explicit Euler integrations; using operator splitting, these can be composed into a consistent numerical integrator for linear time-varying systems. Then, it is shown that the composed integrator is precisely equivalent to SGDm when the integration time step is $\sqrt{\eta}$. $\square$

The first order consistency guarantee in Proposition 2 means that when the step-size $\eta$ is small and initial conditions agree between continuous and discrete time, i.e. $\theta_0 = \theta(0)$, the continuous and discrete time trajectories approximate each other as $\theta_k \approx \theta(k\sqrt{\eta})$, where the difference between them depends on the integration time step $h = \sqrt{\eta}$ (Hairer et al., 2006). Therefore, assuming $\theta_0 = \theta(0)$, the difference accrued in one step of $k$ (local error) is $\theta_1 = \theta(\sqrt{\eta}) + O(\eta)$ and over arbitrarily many steps (global error) is $\theta_k = \theta(\sqrt{\eta}k) + O(\eta k)$.

### 3.3 Parametric Resonance for ODE Convergence and Divergence

The conditions sufficient for convergence and divergence in equation 3 may be shown using established dynamical systems theory. The conditions sufficient for divergence are precisely the conditions for parametric resonance, when $B(t)$ is periodic. Note that periodicity in the first or second moment of $\{X_k\}$ is sufficient to induce periodicity in $B(t)$. As per elementary results in linear ODE theory, the system in equation 3 can be transformed into a linear time-varying first order form

$$\dot{\xi}(t) = A(t)\xi(t) \qquad\qquad A(t) = \begin{bmatrix} 0_{d\times d} & I_{d\times d} \\ B(t) & \frac{1-\mu}{\sqrt{\eta}}I_{d\times d} \end{bmatrix} \qquad (4)$$

where solution trajectories $\theta(t)$ of equation 3 are embedded in solution trajectories $\xi(t)$ of equation 4. Equation 4 admits a fundamental solution matrix[1] $\psi(t)$ such that the spectral radius $\rho$ of $\psi(T)$ characterizes ODE instability, which implies divergence for SGDm, in Theorem 1.

**Theorem 1.** *When $B(t)$ is periodic such that $B(t) = B(t + T)$ for some $T > 0$, the spectral radius $\rho$ of $\psi(T)$ characterizes the stability of solution trajectories of equation 3 as follows:*

- $\rho > 1 \implies$ *trivial solution $\theta(t) = \theta^*$ is unstable. All other solutions diverge as $\theta(t) \to \infty$ exponentially with rate $\rho$.*
- $\rho < 1 \implies$ *trivial solution is asymptotically stable, all other solutions converge as $\theta(t) \to \theta^*$ exponentially with rate $\rho$.*

*Proof Sketch:* ODE 3's stability depends on the relationship between the coefficient $\frac{1-\mu}{\sqrt{\eta}}$ and the expected gradient signal because they determine the spectral radius of $\psi(T)$. The spectral radius $\rho$ is the convergence/divergence rate per period $T$ towards/away from the stationary point $\theta^*$. $\square$

The spectral radius conditions characterize when ODE solution trajectories $\theta(t)$ will converge or diverge, and Proposition 2 tells us that these solution trajectories are approximations of discrete time SGDm trajectories $\{\theta_k\}$ under identical initialization. But does convergence or divergence of $\theta(t)$ imply the same for SGDm? Experiment 4.1 empirically suggests that the approximation is sufficient, since the boundary at $\rho = 1$ in Figure 2a agrees with both continuous and discrete time. However, given Theorem 1 and Proposition 2, we have a theoretical guarantee of SGDm's divergence, but not its convergence. This is because the divergence rate of $\theta(t)$ is exponential, and Proposition 2 provides a bound on long tail behaviour as $\theta_k = \theta(\sqrt{\eta}k) + O(\eta k)$. Since the approximation error is linear in time $k$, and the divergence rate is exponential, the divergence rate dominates, and we are guaranteed that a diverging $\theta(t)$ corresponds to a diverging $\{\theta_k\}$. However, the same argument does not imply that a convergent $\theta(t)$ corresponds to a convergent $\{\theta_k\}$, because the linear error bound technically permits the discrete trajectory to escape $\theta^*$ at a linear rate. Empirically, we see agreement for both convergent and divergent cases, but we defer theoretical proof to future work.

**Summary:** There is a chain of dependencies starting from the non-iid sampling in $\{X_k\}$, and ending at the monodromy's spectral radius.

$$\{X_k\} \to X(t) \to B(t) \to A(t) \to \psi(t) \to \psi(T) \to \rho$$

In order, we have the discrete stochastic process $\{X_k\}$ from which training inputs are sampled, and its underlying continuous stochastic process $X(t)$. If $\{X_k\}$ and $X(t)$ were iid, then the matrix $B$

---

[1] A fundamental solution matrix for system equation 4 is any matrix-valued function of time $\psi(t)$ whose columns are linearly independent solutions to equation 4. We choose $\psi(t)$ such that $\psi(0) = I_{2d\times 2d}$.

would be constant, but the non-iid nature means $B(t)$ is a function of time. The matrix $A(t)$ is the matrix describing SGDm's continuous time dynamics as a linear time-varying ODE $\dot{\xi} = A(t)\xi$, and the matrix $B(t)$ is a simply a submatrix of $A(t)$. Since we have a linear ODE, the space of solution trajectories is spanned by the columns of the ODE's fundamental solution matrix $\psi(t)$. Moreover, since we have a periodic $A(t)$ with period $T$, the stability of all solutions can be determined simply from the largest eigenvalue of $\psi(T)$, a.k.a. its spectral radius $\rho$.

The intuition behind $\rho$'s importance comes from the following fact: after each elapsed period $T$, the phase space (including weight space) is subject to the linear transformation $\psi(T)$. So as time increases, say $n$ periods, the linear transformation is applied iteratively as $\psi(T)^n$, so it is clear that any eigenvalue larger than unity means all solutions eventually diverge exponentially. This is precisely the parametric resonance condition. For a detailed example of all these quantities characterizing the divergence of a simple learning system, see Appendix A.1.

**Technical Novelty:** There are two key novel technical contributions to obtain the results above. (1) The ODE correspondence for non-iid data in combination with momentum methods is new. The non-iid data, which induces time variation in ODE dynamics, is subtle to address, because elementary ODE correspondence (i.e. consistency guarantees) require time invariant ODE dynamics. Our proof specifically addresses this difficulty using operator splitting theory from recent numerical integration literature. (2) Once we have this ODE, it is straightforward to recognize that it is a parametric oscillator. The utility of this connection, however, is that it opens up entirely new avenues for analysis, since there is deeply established literature on the dynamics of these systems. The primary second technical novelty, therefore, is to leverage this connection, and use Floquet theory to give precise divergence conditions for a given $\eta$ and $\mu$, under periodic covariate shift.

## 4 VALIDATING THEORY & ABLATING TOWARDS CONDITIONS IN THE WILD

We now empirically evaluate the effect of resonance in learning problems. First, Experiment 4.1 validates the theoretical predictions by investigating the dynamics of a learning problem which closely matches the theoretical assumptions in Section 3. Specifically, we assess whether or not the spectral radius $\rho$ predicts optimizer convergence or divergence. Then, the remaining Experiments 4.2 - 4.6 test for this phenomena in settings beyond our theory. Each experiment makes a single step away from a theoretical assumption, first relaxing the periodicity assumption, then using stochastic rather than expected gradients and then moving to nonlinear models (neural networks) and optimizers (ADAM).

Across all experiments, input samples at each training step $k$ are drawn from Gaussian distributions with diagonal covariance matrices, i.e. $X_k \sim \mathcal{N}(\bar{x}_k, cI_{d \times d})$, and we induce covariate shift by constructing a time-varying mean sequence $\{\bar{x}_k\}$. Each mean sequence is designed such that (a) its frequency content can be swept with a single parameter $f$ or $T$ and (b) iid sampling is induced by setting $f = 0$ or $T = 0$, so that each experiment has an iid baseline for comparison. The specification of $\bar{x}_k$ is provided in Table 1 for each experiment. This empirical design allows us to systematically test response to different frequencies. See Appendix A.5 for specification of all experiment details, including visual depictions of all synthetic $\{X_k\}$ and their frequency content.

### 4.1 VALIDATING THEORY

We start in a setting as close as possible to the theoretical predictions, with linear regression for a quadratic loss, and covariate shift such that the mean $\mathbb{E}[X_k]$ varies as a strict sinusoid. We perform regression in two weights (i.e. inputs $X_k$ and labels $Y_k$ are scalar-valued). We draw 20 samples from each $X_k$, so that the loss gradient for each time step $k$ is close to its expected value.

Since we have only two weights, the learning system's underlying ODE can be easily specified, such that the fundamental solution matrix can be numerically computed. As per Theorem 1, we can use the spectral radius of the fundamental solution matrix evaluated at time $T$ to make theoretical predictions of where the system should converge or diverge. As depicted in Figure 2a, the theory agrees very well with empirical results. This suggests that parametric resonance is indeed the dominant mechanism behind SGDm divergence under covariate shift. In the appendix, refer to example A.1 for the procedure used to compute the theoretical predictions (i.e. spectral radii $\rho$), and Figure 6a for the full surface from which the contour lines in Figure 2a are rendered.

Table 1: Covariate shift details for each experiment.

| Experiment | Sweep Param. | Mean Sequence $\{\bar{x}_k\}_{k\in\mathbb{N}}$ |
|---|---|---|
| 4.1 Validating Theory | $f \in [0, 0.05]$ | $\bar{x}_k = 0.5\sin(2\pi f k)\,,\, T = 1/f$ |
| 4.2 Ablating Periodicity | $f \in [0, 0.05]$ | $\bar{x}_k = \phi_1\bar{x}_{k-1} + \phi_2\bar{x}_{k-2} + \xi_k$ 
 $\phi_1 = 4\phi_2(\phi_2 - 1)^{-1}\cos(2\pi f)$ 
 $\phi_2, \xi_k, \bar{x}_1, \bar{x}_2$ see Table 3 |
| 4.3 Ablating Expected Gradient | $T \in [0, 120]$ | $\bar{x}_k = \begin{cases} \frac{\xi}{2\|\xi\|} & \text{if } \lfloor\frac{2k}{T}\rfloor \equiv 0 \mod 1 \\ \frac{-\xi}{2\|\xi\|} & \text{if } \lfloor\frac{2k}{T}\rfloor \equiv 1 \mod 1 \end{cases}$ 
 $\xi \sim \mathcal{N}(0, I_{d\times d})$ iid |
| 4.4 Ablating Periodicity Further | $T \in [0, 50]$ | $\bar{x}_k = \xi_i$ where $i = \left\lfloor\frac{k}{T}\right\rfloor$ 
 $\xi_i \sim \mathcal{N}(0, vI_{d\times d})$ iid |
| 4.5 Ablating Optimizer Linearity | $T \in [0, 100]$ | $\bar{x}_k$ same as above. |
| 4.6 Ablating Model Linearity | $T \in [0, 100]$ | $\bar{x}_k$ same as above. |

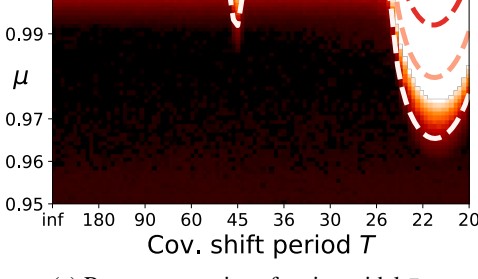

(a) Resonance regions for sinusoidal $\bar{x}_k$

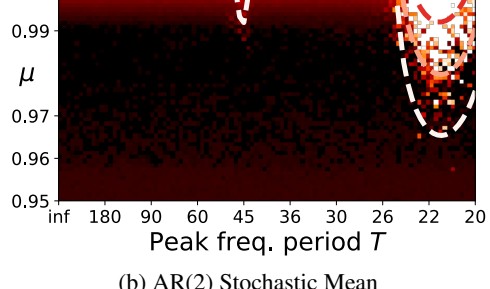

(b) AR(2) Stochastic Mean

Figure 2: Empirical heatmap of momentum $\mu$ versus period $T$ for SGDm for linear regression, overlaid by contours of theoretical prediction. Each pixel is the distance $\|\theta_k - \theta^*\|$ averaged over the final 500 steps $k$ and 10 runs. Dark pixels converge quickly and stably, bright pixels diverge exponentially. The contours show divergence predictions from Theorem 1: the white contour has $\rho = 1$, with $\rho$ increasing with redness.

## 4.2 ABLATING PERIODICITY

We repeat Experiment 4.1, but with the mean of $X_k$ varying stochastically instead of deterministically. The result in Theorem 1 assumes the expected gradient varies periodically over time given fixed weights $\theta$. But even with aperiodic and/or stochastic time variation, our LTV system equation 4 might be similarly susceptible to instability; the theory does not have periodicity as a necessary condition of divergence. To test this, we replace our periodic covariate shift mean with a mean that moves according to an AR(2) process. The AR(2) process is tuned to have a frequency peak exactly at the frequency of the sinusoidal covariate shift of Experiment 4.1, to change only the nature of the shift—aperiodicity—rather than the frequency.

We can see from the heatmap in Figure 2b that under this setting we observe very similar resonance behaviour in the learning system, which aligns well with the identical predicted stability regions. Note that both theoretical and empirical results in Figures 2a and 2b suggest that sufficiently low momentum values $\mu$ mitigate the resonance phenomenon, which agrees with the role it plays in the ODE system: decreasing $\mu$ increases damping. We also observe that resonance regions shrink as step size $\eta$ is decreased; see the Appendix for plots demonstrating this trend. Analytically characterizing the bounds of stable $\mu, \eta$ in terms of system properties is an interesting future direction.

### 4.3 ABLATING EXPECTED GRADIENT

Here we perform linear regression with periodic mean covariate shift, and ablate the number of samples drawn from each $X_k$ to show the effect of increasing noise in the gradient signal. Linear regression is performed mapping $\mathbb{R}^5 \to \mathbb{R}$, with the weights learned via SGDm. The mean sequence $\bar{x}_k$ oscillates with strict periodicity between $\pm\bar{x}$, where $\bar{x}$ is a unit norm 5-vector randomly chosen for each run. i.e. the mean signal is a square wave in $\mathbb{R}^5$ with period $T$.

Linear regression implies a quadratic loss, and periodic covariate shift implies a loss with periodic time variation in expectation, so we are very near to the setting in which Theorem 1 is directly applicable. However, as we ablate from 5 samples to 1 sample drawn from each $X_k$, we move further away from the expected gradient, towards the fully stochastic gradient setting. As we can see in Figure 3 resonance is dampened by stochasticity in the gradient signal, but nonetheless is still present. These findings are further supported in Appendix A.3, where Experiments 4.1 and 4.2 are also ablated from expected to stochastic gradient signals.

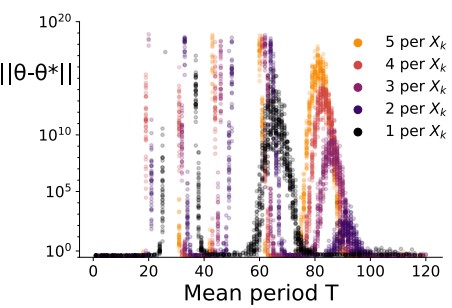

Figure 3: Regression w/ periodic $\bar{x}_k$. Each dot is avg. dist. $||\theta_k - \theta^*||$ over final 500 steps. Each color has three peaks: the left peak exceeds y scale for all curves, the center peak is small enough to appear only for the black curve, and the right peak appears for all curves (vanishingly small for the black curve.) Resonance is dampened and right-shifted by decreasing samples per $X_k$.

### 4.4 ABLATING PERIODICITY FURTHER

We further depart from strictly periodic covariate shift by having piece-wise stationary means and non-smooth changes: we randomly sample a new mean from a normal distribution $\mathcal{N}(0, v^2)$ after every $T$ update steps. We further investigate the impact of this covariate shift under (a) increasing variance $v^2$ and (b) increasing dimensionality of the inputs.

We observe divergence characteristic of the parametric resonance, with smaller $T$ having a highly divergent response. In Figure 4a we can see that a higher variance results in much more instability, though even for smaller variance the distance to $\theta^*$ is still on the order of $10^3$. This results makes sense, as the variance is connected to the driving signal amplitude, on which parametric resonance has a strong dependence. We also observe a strong dependence on the number of input space dimensions, in Figure 4b. This aligns with the fact that the expected norm of samples drawn from a multivariate Gaussian increases with dimensionality, even with a fixed covariance scale (of 0.1). This result suggests that resonance phenomena may actually be exacerbated for higher-dimensional inputs, which is the setting we expect to see in practice.

### 4.5 ABLATING OPTIMIZER LINEARITY

So far our experiments have ablated periodicity, input dimensionality, and gradient stochasticity. As per the conditions in Section 3, each of these ablated systems still corresponds to a discretization of an LTV ODE. We now ask whether resonance can be observed in systems which do not correspond to a discretization of our ODE. To investigate this hypothesis, we replace SGDm with ADAM as an update rule for our learning algorithm. We no longer have the ODE representation of this learning system, but can empirically investigate the behaviour of the system under covariate shift.

We use the same stochastic mean switching problem as the previous experiment, where we regress from 5 dimensions to 1 with input samples. Again, we measure the distance $||\theta_k - \theta^*||$, and we vary the ADAM parameter $\beta_1$. Similar to the SGDm optimizer, we can see in Figure 4c that there is a band of mean switching intervals $T$ for which convergence is worse.

Unlike SGDm, there is no divergence. Proper parametric resonance induces exponential divergence, which is why the previous results were presented with Euclidean distances between weights on the order of $10^{19}$. Note that in Figure 4c, the frequency response in weight space distance is instead measured on the order of $10^0$.

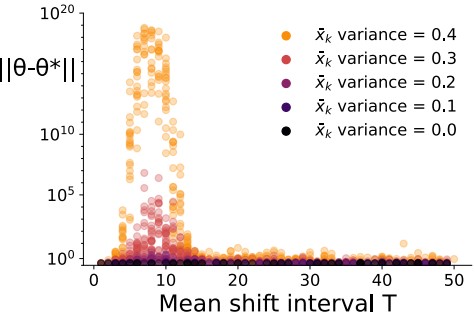

(a) SGDm w/ stochastic $\bar{x}_k$, variance sensitivity

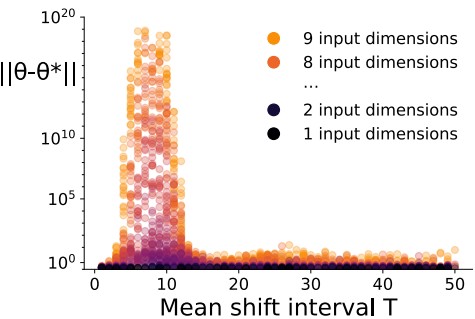

(b) SGD w/ stochastic $\bar{x}_k$, sensitivity to $d$

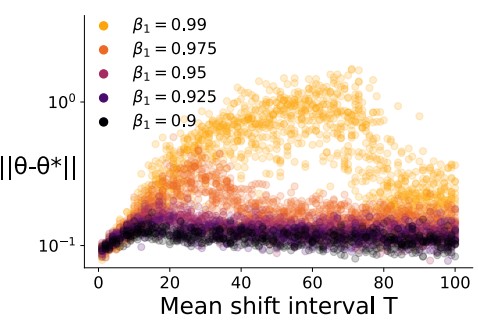

(c) ADAM w/ stochastic $\bar{x}_k$, $\beta_1$ sensitivity

Figure 4: Regression w/ stochastic $\bar{x}_k$. Each marker is avg. distance $||\theta_k - \theta^*||$ over final 500 steps (a and b) or 2000 steps (c). For SGDm, resonance is very sensitive to the $\bar{x}_k$ signal variance (i.e. amplitude) in (a), and to the number of input dimensions in (b). For ADAM in (c), convergence is affected by specific frequency content (x-axis) but does not diverge, suggesting that the frequency response is significantly damped. Similar to SGDm, response is higher for larger momentum parameters $\beta_1$.

## 4.6 ABLATING MODEL LINEARITY

Our final step away from ODE equation 3 abandons the quadratic loss surface. We fit a nonlinear target function using a fully connected ReLU network, reusing the previous covariate shift. In previous experiments, we measured the distance $||\theta_k - \theta^*||$. Now a unique target is not guaranteed, so frequency response is in terms of loss. In order to amplify frequency response, 10 samples were drawn from each $X_k$ to reduce gradient signal noise.

Figure 5 shows a band of mean switching intervals $T$ which damage convergence, which is characteristic of resonance. Though we do not see divergence, the impact on accuracy is problematic. In this case, models with loss $> 0.3$ fit extremely poorly, but those with $< 0.05$ perform well.

## 5 DISCUSSION

This work is a first step towards understanding frequency response due to covariate shift. Here we discuss future directions, limitations, and practicality.

**Limitations and Next Steps:** In our theoretical analysis, we take expectation over observation noise and over the distribution of inputs $X_k$, so the gradient coefficient matrix $B(t)$ in the ODE equation 3 is deterministic. This allows rigorous characterization

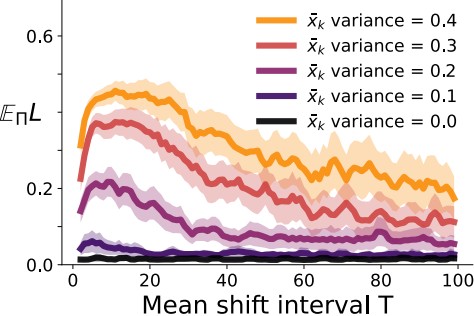

Figure 5: Training a neural network with SGDm shows a peak response in the loss around the band $T \in [5, 40]$. The y-axis is average test loss over the final 2000 training steps over 20 runs, with test set obtained via the stationary distribution of $\{X_k\}$. Shaded regions are 95% confidence intervals.

of resonance for periodic $B(t)$. Experiments 4.2 and 4.4 demonstrate that resonance still drives divergence when $B(t)$ is nondeterministic, so rigorous exploration in future work might employ stochastic differential equations. In particular, Experiment 4.2 aligns exactly with the theoretical predictions, suggesting that the resonance is in response to the frequency content of $X_t$, not its periodicity.

For the sake of consistency across experiments, our non-iid sampling is always induced by Gaussian $X_k$ with fixed diagonal covariance matrix. But Assumption 1 is far more relaxed, suggesting resonance is possible with time-varying covariance and non-Gaussian distributions, which may be explored in future empirical work

When assessing a new phenomenon like covariate shift resonance, assessment on simple problems is a critical first step, and we have endeavored to be comprehensive in that assessment. But we only scratch the surface of more complex optimizers and nonlinear models. Similarly, for the sake of experimental control and interpretability, we assess resonance on synthetic data instead of real data. While it is obvious that many sources of real data have nontrivial frequency content (e.g. audio samples, machine sensors, etc.) it is not yet clear when realistic frequency content will resonate with the system trying to learn from it. Optimizers, models, and data all pose exciting future directions for this work: to extend into the frequency *response* of more complex optimizers and models, and into the frequency *content* of real data.

Finally, the theory in this work is for linear models. This is the correct setting to understand first, and is of independent interest. Our theory applies to any linear models that are augmented with a nonlinear basis (such as a Fourier basis, polynomials or radial basis functions) by examining the frequency content of the basis output signal.

**Practical Implications:** One potential conclusion from our results is that though resonance could potentially arise in our learning systems, it may not be problematic. First, the heatmaps in Figures 2a and 2b indicate that SGDm converges much more often than it diverges, and that we can practically avoid resonance simply by reducing the momentum parameter $\mu$. As per Appendix A.2, we find that sufficiently low $\mu$ does mitigate resonance. Second, we may simply be able to avoid covariate shift in the first place, by keeping buffers of data and shuffling samples.

However, the reality is not so simple. Higher levels of momentum are typically more effective in practice. There is a tension between the level of momentum and avoiding resonance, that cannot easily be addressed by simply picking low momentum by default. Instead, smarter approaches which recognize potential resonance and then reduce momentum may be a much more effective way to guard against this issue; the development of such methods can leverage the insights in this work. In addition, relying on well-behaved or shuffled inputs may be impractical in certain settings. Algorithms in signal processing and the streaming literature often process data sequentially. Reinforcement learning approaches use sliding window buffers (in replay), where covariate shift may still persist across buffers. Further, it is even possible that adversarial attacks could be designed to exploit this weakness of SGDm, causing poor performance or even instability from attackers attempting to generate data that results in resonant behavior.

Our work also suggests avenues towards the development of diagnostic tools. Indeed, our experimental approach to investigate this phenomena mimics those in the modal analysis literature (Avitabile, 2017). We treat the input sampling process as an input signal which is noisy and/or stochastic, vary the signal's frequency content across a wide band, and measure the learning system's output as a response across input frequencies. This is analogous to a mechanical engineer triggering an oscillatory vibration within a machine's engine compartment, sweeping across frequencies, and measuring the resulting vibration amplitude in the machine's housing.

## 6 CONCLUSION

To our knowledge, this work is the first to investigate SGDm under covariate shift from the ODE perspective. This perspective reveals new insights, in particular the revelation that SGDm under covariate shift numerically integrates a parametric oscillator, rather than the simple undriven harmonic oscillator induced by iid sampling. We leverage dynamical systems theory to analyze the stability of a learning system using SGDm under covariate shift, and are able to provide conditions for exponential divergence due entirely to the frequency content induced by this non-iid sampling. Despite its age and establishment, understanding SGDm is an ongoing process. We contribute to this process in a way that provides physical intuition, rather than a merely algebraic proof. We hope that the connection we have drawn to parametric oscillation in dynamical systems theory will help to further propel understanding of non-iid sampling and SGDm.

## 7 ACKNOWLEDGEMENTS

We would like to thank Yurij Salmaniw and Ryan Thiessen for their discussions regarding Floquet theory and helping us makes sense of our differential equations. We would also like to thank Csaba Szepesvari for pointing us to excellent resources for numerical integration, algorithmic convergence, stochastic processes, and various future directions for this work.

## 8 ETHICS STATEMENT

Our work is theoretical and attempts to provide greater understanding of an optimization algorithm. Due to the remoteness of this work from direct ethical concerns, it is difficult to reason about any ethical implications of the work. We hope that the insights into SGDm could help to make machine learning algorithm more robust, but beyond that we are not aware of any ethical concerns or implications of this work.

## 9 REPRODUCIBILITY STATEMENT

We have endeavored to be very clear and thorough in our explanation of the theoretical results, and all assumptions and proofs are explicitly stated and contained in either the main body of the paper or in the appendix. The code for all of the experiments is included in the supplementary material and the instructions for reproduction are in the README file. The data used is synthetic and code to generate the data is contained in the supplementary material. All of the experimental details (e.g. hyperparameter selection) are contained in tables in the paper and appendix and the supplementary material. All of the experiments should be easily run on modest commodity hardware in the space of a few days, at most.

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

# A   APPENDIX

## A.1   A PROCEDURE TO NUMERICALLY DETERMINE SYSTEM STABILITY

Here we expand on the example given at the end of Section 3. As a reminder, we consider online linear least squares regression using SGDm with a fixed $\eta, \mu$ under a periodic covariate shift. Data is sampled from the sequence of random variables $\{X_k, Y_k\}_{k \in \mathbb{N}}$ where $X_k$ is normally distributed and shifts periodically in expectation. $Y_k$ is an unchanging linear function of $X_k$ with zero-mean observation noise. For ease of exposition we consider the one dimensional regression problem with $X_k \in \mathbb{R}$ and $Y_k \in \mathbb{R}$, however the theory applies to any learning system representable as an LTV ODE satisfying the assumptions required for Theorem 1.

Under this covaiate shift, $\bar{x}_k$ varies sinusoidally in $[-a, a]$ with period $T = \frac{1}{f}$. Note that although $X_k$ is periodic in expectation it admits a stationary distribution. The full specification of the data generating process is as follows

$$
\begin{aligned}
X_k &\sim \mathcal{N}(\bar{x}_k, 1) \\
\bar{x}_k &= a \cos(2\pi f k) \\
Y_k &= \theta_1^* X_k + \theta_2^* + \epsilon \\
\epsilon &\sim \mathcal{N}(0, 1)
\end{aligned}
$$

where $\theta^* = [\theta_1^*, \theta_2^*]^T$ are the target weights.

We consider a squared error loss function,

$$
\begin{aligned}
L(z; \theta) &= (\hat{y} - y)^2 \\
&= [\langle \theta, \mathbf{x} \rangle - (\langle \theta^*, \mathbf{x} \rangle + \epsilon)]^2 \\
&= [(\theta_1 x + \theta_2) - (\theta_1^* x + \theta_2^* + \epsilon)]^2
\end{aligned}
$$

Hence, taking gradients at time $k$ w.r.t. $\theta_k$ we have

$$
\nabla_{\theta_k} L(z_k; \theta_k) = \begin{bmatrix} 2x_k[(\theta_{k_1} x_k + \theta_{k_2}) - (\theta_{k_1}^* x_k + \theta_{k_2}^* + \epsilon)] \\ 2[(\theta_{k_1} x_k + \theta_{k_2}) - (\theta_{k_1}^* x_k + \theta_{k_2}^* + \epsilon)] \end{bmatrix}
$$

Taking expected gradients,

$$
\begin{aligned}
\mathbb{E}[\nabla_{\theta_k} L(z_k; \theta_k)] &= \begin{bmatrix} \mathbb{E}[2[(\theta_{k_1} x_k^2 + \theta_{k_2} x_k) - (\theta_{k_1}^* x_k^2 + \theta_{k_2}^* x_k + \epsilon x_k)]] \\ \mathbb{E}[2[(\theta_{k_1} x_k + \theta_{k_2}) - (\theta_{k_1}^* x_k + \theta_{k_2}^* + \epsilon)]] \end{bmatrix} \\
&= \begin{bmatrix} 2[(\theta_{k_1}(1 + \bar{x}_k^2) + \theta_{k_2} \bar{x}_k) - (\theta_{k_1}^*(1 + \bar{x}_k^2) + \theta_{k_2}^* \bar{x}_k + \mathbb{E}[\epsilon] \bar{x}_k)] \\ 2[(\theta_{k_1} \bar{x}_k + \theta_{k_2}) - (\theta_{k_1}^* \bar{x}_k + \theta_{k_2}^* + \mathbb{E}[\epsilon])] \end{bmatrix} \\
&= 2 \begin{bmatrix} (\theta_{k_1}(1 + \bar{x}_k^2) + \theta_{k_2} \bar{x}_k) - (\theta_{k_1}^*(1 + \bar{x}_k^2) + \theta_{k_2}^* \bar{x}_k) \\ (\theta_{k_1} \bar{x}_k + \theta_{k_2}) - (\theta_{k_1}^* \bar{x}_k + \theta_{k_2}^*) \end{bmatrix} \\
&= 2 \begin{bmatrix} \theta_{k_1}(1 + \bar{x}_k^2) - \theta_{k_1}^*(1 + \bar{x}_k^2) + \theta_{k_2} \bar{x}_k - \theta_{k_2}^* \bar{x}_k \\ \theta_{k_1} \bar{x}_k - \theta_{k_1}^* \bar{x}_k + \theta_{k_2} - \theta_{k_2}^* \end{bmatrix} \\
&= 2 \begin{bmatrix} (1 + \bar{x}_k^2) & \bar{x}_k \\ \bar{x}_k & 1 \end{bmatrix} \begin{bmatrix} \theta_{k_1} - \theta_{k_1}^* \\ \theta_{k_2} - \theta_{k_2}^* \end{bmatrix}
\end{aligned}
$$

In the notation of Proposition 1, we can write this as

$$
g_k(\theta_k) = B_k[\theta_k - \theta^*]^T \qquad \text{with} \quad B_k = 2 \begin{bmatrix} (1 + \bar{x}_k^2) & \bar{x}_k \\ \bar{x}_k & 1 \end{bmatrix}
$$

If we let $\bar{x}(t) = a \cos(2\pi f \eta^{-\frac{1}{2}} t)$, then we induce an $X(t)$ from which $\{X_k\}$ is sampled, as per Section A.4.3. This results in $B_k = B(\sqrt{\eta} k$ with the matrix-valued function $B(t)$

$$
B(t) = 2 \begin{bmatrix} (1 + \bar{x}(t)^2) & \bar{x}(t) \\ \bar{x}(t) & 1 \end{bmatrix}
$$

From Proposition 2, we have that our learning system numerically integrates an LTV of the form

$$\ddot{\theta}(t) + \frac{1-\mu}{\sqrt{\eta}}\dot{\theta}(t) + B(t)(\theta(t) - \theta^*) = 0 \qquad \text{with} \quad (\theta(t) - \theta^*) = \begin{bmatrix} \theta_1(t) - \theta_1^*(t) \\ \theta_2(t) - \theta_2^*(t) \end{bmatrix} \qquad (5)$$

Note that $B(t)$ is piecewise continuous and periodic with period $T = \frac{1}{f}$.

This second order ODE can be represented as a system of first order ODEs with the transformation

$$\xi_1 = \theta_1 - \theta_1^*$$
$$\xi_2 = \theta_2 - \theta_2^*$$
$$\xi_3 = \dot{\theta}_1$$
$$\xi_4 = \dot{\theta}_2$$

so that the equation equation 5 can be written in standard first order linear form $\dot{\xi} = A(t)\xi$:

$$\begin{bmatrix} \dot{\xi}_1 \\ \dot{\xi}_2 \\ \dot{\xi}_3 \\ \dot{\xi}_4 \end{bmatrix} = \begin{bmatrix} 0 & 0 & 1 & 0 \\ 0 & 0 & 0 & 1 \\ -2(1 + \bar{x}^2(t)) & -2\bar{x}(t) & \frac{\mu-1}{\sqrt{\eta}} & 0 \\ -2\bar{x}(t) & -2 & 0 & \frac{\mu-1}{\sqrt{\eta}} \end{bmatrix} \begin{bmatrix} \xi_1 \\ \xi_2 \\ \xi_3 \\ \xi_4 \end{bmatrix} \qquad (6)$$

The matrix $B(t)$ is piecewise continuous and periodic, which implies the matrix $A(t)$ also shares these properties. The ODE equation 5 satisfies all the assumptions needed to make use of Theorem 1 to determine the stability of solution trajectories. We can obtain a fundamental solution matrix, $\psi(t)$, of equation 6 satisfying initial conditions $\psi(0) = I_{4\times4}$ by numerical computation through the use of an ODE solver.

Once we have this matrix $\psi(t)$, we evaluate it at time $T = \frac{1}{f}$ to obtain the system's monodromy matrix. Now we only need to determine the spectral radius, $\rho$, of $\psi(T)$ to determine the stability of solution trajectories for the system. Details regarding why this is true are given in the proof of Theorem 1.

We repeat this process, sweeping over values of $\mu$, $\eta$, and $f$ to determine the stability of solution trajectories for systems to triples of fixed values for these hyperparameters. The results of this process are given in Figure 6 below.

## A.2 Reducing Resonant Responses

In experiments 4.1 and 4.2, we see that as the momentum parameter is reduced, the tendency to resonate is mitigated. This aligns with theoretical predictions, which (in the context of those particular experiments) suggest that a sufficiently low $\mu$ makes SGDm convergent across all frequencies in the band we evaluate. Intuitively, this aligns with the role $\mu$ plays in the ODE, as it appears in the following coefficient on the first order derivative of system equation 3 (i.e. the system's 'damping' coefficient), repeated below with the damping coefficient $\alpha$ explicitly labelled:

$$\ddot{\theta}(t) + \alpha\dot{\theta}(t) + B(t)(\theta(t) - \theta^*) = 0 \qquad\qquad \alpha := \frac{1-\mu}{\sqrt{\eta}} \qquad (7)$$

When $\alpha = 0$, the system has no damping (i.e. friction is zero, in the physical analogue) so resonant responses are maximized. As $\alpha$ increases, damping increases, and resonant responses are reduced. There are two ways to increase $\alpha$: reduce $\mu$ or reduce $\eta$.

### A.2.1 Reducing the momentum parameter

Figure 6b shows the theoretical heatmap across all possible momentum values, and across a much wider band of frequencies, which suggests a trend: setting $\mu$ to a sufficiently low value will completely mitigate resonance, though setting $\mu$ too low will worsen convergence rate. For now, we will set aside the observation that $\mu$ too low worsens convergence rate, and we will now show that reducing $\mu$ will reliably dampen resonance across all other experiments.

In 4.5, ADAM's parameter $\beta_1$ was varied, and we see that reducing $\beta_1$ decreases the tendency to resonate. Since $\beta_1$ is the nearest parameter to $\mu$ in SGDm, the desired trend has already been

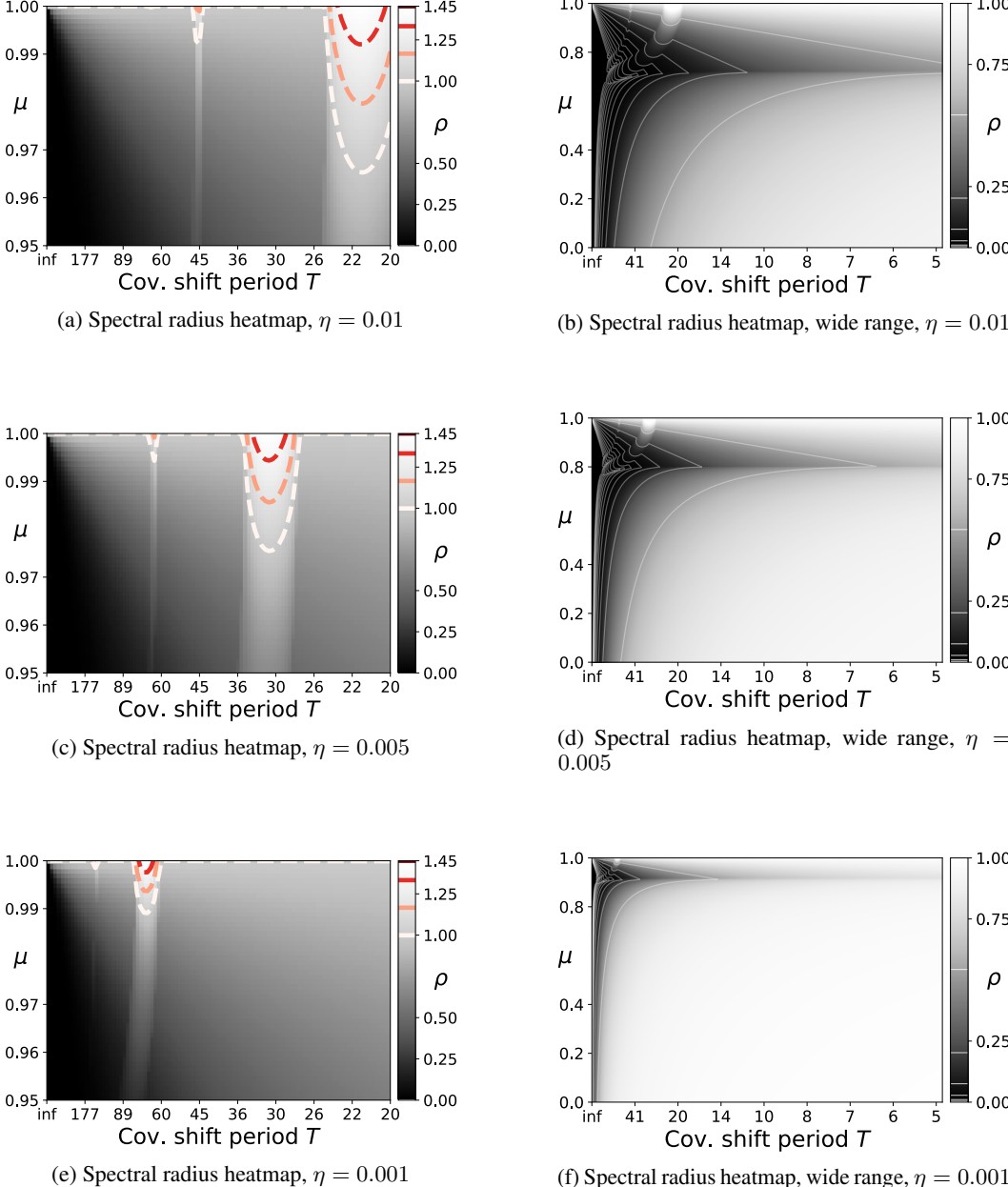

Figure 6: Spectral radii of the monodromy matrices induced by particular momentum $\mu$ and period $T$ values (x and y pixel coordinates, respectively). Step-size $\eta$ decreasing with each row. Each column shares the same range of $\mu, T$. The right column shows the full range of momentum $\mu \in [0, 1]$ and a much wider range of periods $T$ than the left. The left column figures simply 'zoom in' to the upper left corner of the figure to its right. (a) corresponds to the $\mu, \eta, T$ as Experiments 4.1 and 4.2, with contour lines identical to 2a 2b such that the white line separates the convergent region below from the divergent regions above. (b), with the extent of (a) indicated by dotted lines. The figures in the right column show a horizontal band of minimum spectral radius (i.e. max convergence rate), which suggests that there exists a least-resonating $\mu_{\text{best}}$ across nearly all frequencies, with only minor deviations toward the far left, where data approaches iid. Though the location of $\mu_{\text{best}}$ clearly changes with step-size $\eta$, and likely depends upon other aspects of the specific learning problem.

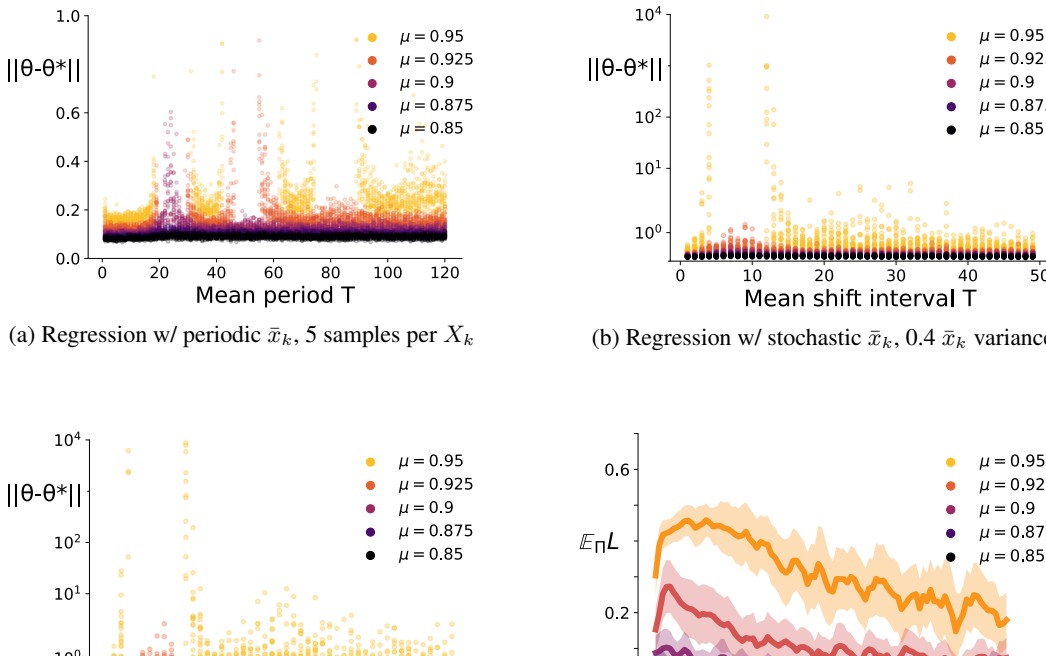

(a) Regression w/ periodic $\bar{x}_k$, 5 samples per $X_k$

(b) Regression w/ stochastic $\bar{x}_k$, 0.4 $\bar{x}_k$ variance

(c) Regression w/ stochastic $\bar{x}_k$, $d = 9$

(d) Neural net w/ stochastic $\bar{x}_k$, $\bar{x}_k$ variance $= 0.4$

Figure 7: Re-running with reduced momentum values for highest resonance configurations chosen from Experiments 4.3 (a), 4.4 (b, c), and 4.6 (d). In all cases, reducing momentum significantly dampens resonant response, with $\mu = 0.85$ completely mitigating resonant response.

demonstrated. For the remainder of this section, we show the resonance-damping behaviour of $\mu$ in the remaining experiments: 4.3, 4.4, and 4.6. In particular, from each experiment we choose the configuration which had the highest tendency to resonate, and we modify the experiment by running them with several decreasing values of momentum $\mu$. See Figure 7 for results. In Section 4, these experiments used momentum $\mu = 0.95$, and here we run with $\mu \in [0.85, 0.95]$, with all other experimental parameters identical.

### A.2.2 REDUCING THE STEP-SIZE

Another way to reduce tendency to resonate is suggested by the damping coefficient equation 7: reducing step-size $\eta$. Here we repeat Experiments 4.1 and 4.2, including two smaller step-sizes $\eta$ to empirically demonstrate that trend. Specifically, Figure 8 shows the resonance heatmap and spectral radius contour lines for regression in two weights with covariate shift, identically to Experiments 4.1 and 4.2. The left column shows sinusoidal covariate shift, and the right column the AR(2) covariate shift. Each row corresponds to a fixed step-size $\eta \in \{0.01, 0.005, 0.001\}$, and it is clear that resonant, diverging regions are significantly reduced in size as step-size is decreased, with a narrower band of frequencies diverging, and the minimum momentum $\mu$ required for resonance increasing towards 1. This trend is reflected in both the empirical heatmap results, as well as the theoretically predicted spectral radius contour lines.

### A.3 EXPERIMENTS 4.1 AND 4.2 WITH STOCHASTIC GRADIENTS

Experiment 4.1 1 serves as a simple setting, as close to the theory of Section 3 as possible. In particular, each descent step made at time $k$ uses the gradients induced by 20 samples from each $X_k$ so that the gradient signal is much closer to the expected gradient than the stochastic setting. Experiment 4.2 uses the same technique.

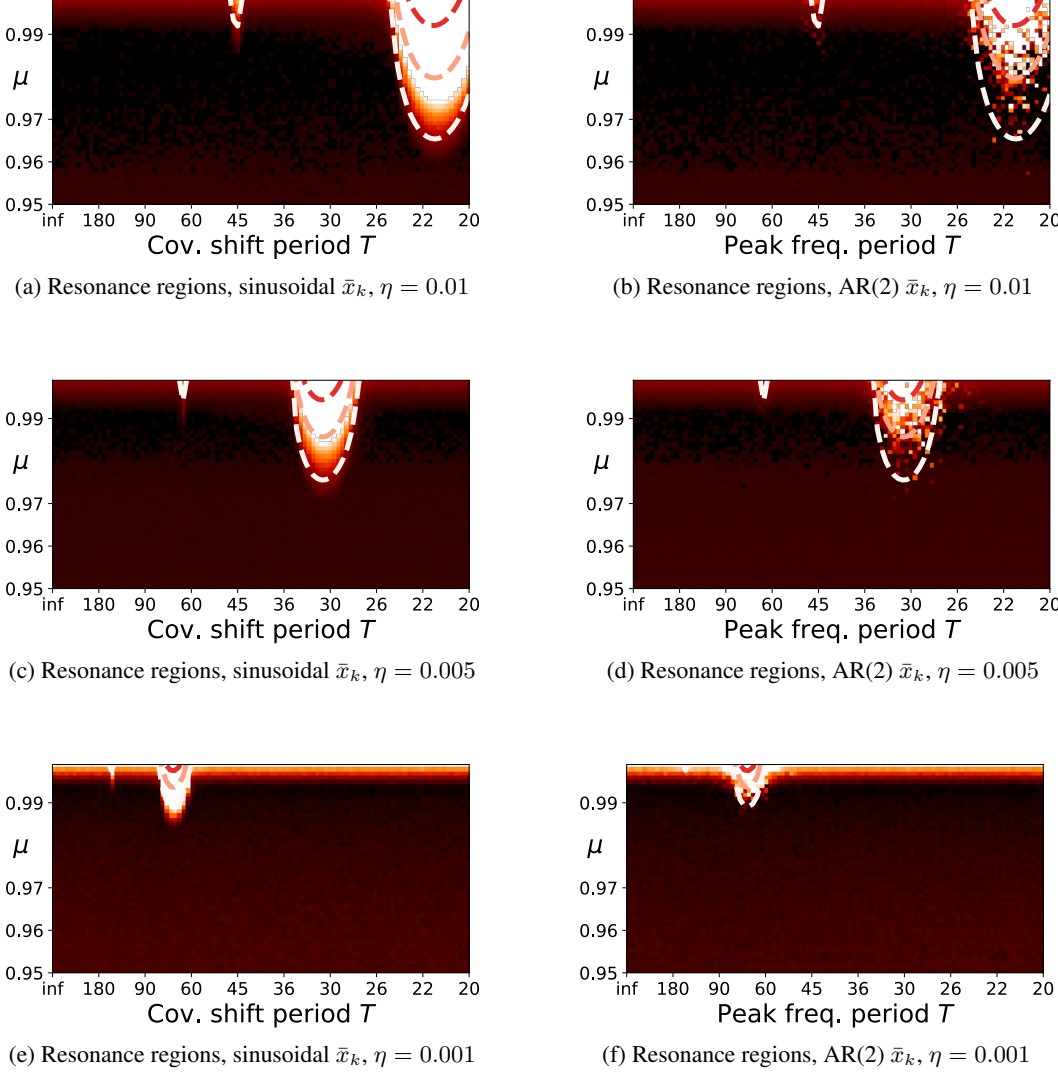

(a) Resonance regions, sinusoidal $\bar{x}_k$, $\eta = 0.01$      (b) Resonance regions, AR(2) $\bar{x}_k$, $\eta = 0.01$

(c) Resonance regions, sinusoidal $\bar{x}_k$, $\eta = 0.005$      (d) Resonance regions, AR(2) $\bar{x}_k$, $\eta = 0.005$

(e) Resonance regions, sinusoidal $\bar{x}_k$, $\eta = 0.001$      (f) Resonance regions, AR(2) $\bar{x}_k$, $\eta = 0.001$

Figure 8: Re-running with reduced step-size for Experiments 4.1 (a, c, e) and 4.2 (b, d, f). In both cases, reducing step-size significantly dampens resonant response.

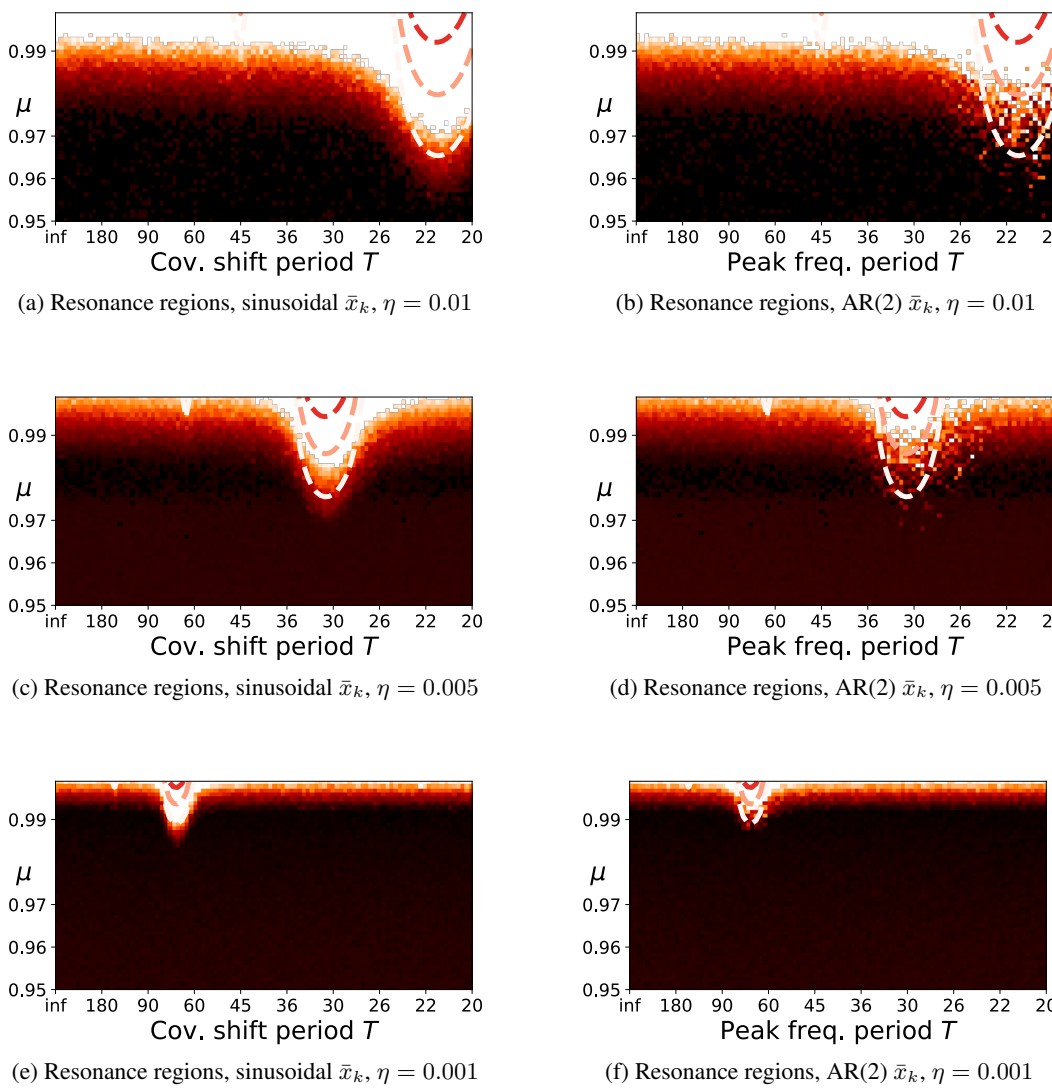

(a) Resonance regions, sinusoidal $\bar{x}_k$, $\eta = 0.01$

(b) Resonance regions, AR(2) $\bar{x}_k$, $\eta = 0.01$

(c) Resonance regions, sinusoidal $\bar{x}_k$, $\eta = 0.005$

(d) Resonance regions, AR(2) $\bar{x}_k$, $\eta = 0.005$

(e) Resonance regions, sinusoidal $\bar{x}_k$, $\eta = 0.001$

(f) Resonance regions, AR(2) $\bar{x}_k$, $\eta = 0.001$

Figure 9: Re-running with fully stochastic gradients for Experiments 4.1 (a, c, e) and 4.2 (b, d, f). In all cases, increasing gradient variance significantly dampens resonant response, but does not change it otherwise.

Here we repeat Experiments 4.1 and 4.2, but drawing only a single sample from each $X_k$, so that the gradient signal is fully stochastic. That is, we significantly increase gradient signal's variance. As Figure 9 shows, the resonant response is significantly damped by increasing gradient signal variance. We observe no other change in the empirical heatmap: regions of instability are neither introduced nor removed, and the existing regions do not change shape or location. Only the size of instability regions is affected, which is most apparent under comparison to Figure 8, where the empirically observed instability regions are expanded to closer match the theoretical predictions derived from expected gradients.

Also the bright band of instability in all subfigures of Figure 9 are wider than those of 8. This aligns with intuition from the conventional setting with iid data, where increasing gradient variance reduces optimizer stability.

## A.4 PROOFS

### A.4.1 ADDITIONAL NOTATION

Wherever we write the training input generating process $\{X_k\}$, it is implicit that the last dimension is fixed at 1 if one wishes to describe linear models with a bias term. In this way, the notation $\langle \theta, X_k \rangle$ accommodates linear models with or without a bias term.

We use $\xi$ to denote the *phase space* coordinates of weights $\theta$, meaning $\xi = \begin{bmatrix} \theta \\ \dot{\theta} \end{bmatrix}$, where vectors $\theta, \dot{\theta} \in \mathbb{R}^d$ are stacked so that the resulting $\xi \in \mathbb{R}^{2d}$. We denote the $d$-dimensional zero vector as $0_d$. For an ODE system $\dot{\xi}(t) = f(\xi(t), t)$ with arbitrary solution trajectory $\xi(t)$ approximated by a sequence of iterates $\{\xi_k\}$, we denote the integration time step as $h$, and the *numerical flow* as $\phi_{h,k}$, which is the map such that $\xi_{k+1} = \phi_{h,k}(\xi_k)$. We denote the $k$-th discrete timestep as $t_k$, which we always use to refer to the $k$-th multiple of integration time step $h$, i.e. $t_k = hk$.

Similar to the main body, when a function of time $\theta(t)$ is approximated by a discrete time sequence $\{\theta_k\}$, they are denoted with the same symbol, and distinguished by parenthetical argument $t$ and subscript $k$, respectively. At the risk of abusing notation, we will adhere to this convention and use $\{\dot{\theta}_k\}$ to be a discrete sequence approximating the function of time $\dot{\theta}(t)$, which refers to the time derivative of $\theta(t)$. This means the symbol $\dot{\theta}_k$ is the $k$-th iterate of a sequence approximating $\dot{\theta}(t)$.

We now restate Assumptions, Propositions, and the Theorem from Section 3, providing complete proofs.

### A.4.2 LINEAR TIME-VARYING EXPECTED GRADIENT VIA COVARIATE SHIFT

**Assumption 1.** *The covariate generating process $\{X_k\}_{k \in \mathbb{N}}$ is not identically distributed:* $Corr(X_{k_1}, X_{k_1}) \neq Corr(X_{k_2}, X_{k_2})$ *for some $k_1, k_2 \in \mathbb{N}$.*

**Assumption 2.** *The targets $Y_k$ are a fixed linear function of $X_k$ with iid zero-mean observation noise, $\epsilon_k$ such that $\mathbb{E}[\epsilon_k] = 0$. That is, $Y_k = \langle \theta^*, X_k \rangle + \epsilon_k$ where $\theta^* \in \mathbb{R}^d$ is fixed for all $k$.*

**Proposition 1.** *Under Assumptions 1, 2, a linear model with weights $\theta_k$ making predictions $\widehat{Y}_k = \langle \theta_k, X_k \rangle$ with a mean squared error (MSE) objective will induce time-varying linear expected loss gradients $g_k(\theta_k) = B_k(\theta_k - \theta^*)$ for all $k \in \mathbb{N}$, where $B_k \in \mathbb{R}^{d \times d}$ and $B_{k_1} \neq B_{k_2}$ for some $k_1, k_2$.*

*Proof.* We will show that the gradient of MSE loss $\nabla_{\theta_k} L(Z; \theta_k)$ is a random variable whose expectation will take the linear form $B_k(\theta_k - \theta^*)$. We start with gradient for arbitrary time step $k$

$$
\begin{aligned}
\nabla_{\theta_k} L(Z; \theta_k) &= \nabla_{\theta_k} (\widehat{Y}_k - Y_k)^2 & \text{MSE def'n} \\
&= \nabla_{\theta_k} [(\langle \theta_k, X_k \rangle - Y_k)^2] & \widehat{Y}_k \text{ def'n from Prop. 1} \\
&= \nabla_{\theta_k} [(\langle \theta_k, X_k \rangle - \langle \theta^*, X_k \rangle + \epsilon_k)^2] & Y_k \text{ def'n from Ass. 2} \\
&= \nabla_{\theta_k} [(\langle \theta_k - \theta^*, X_k \rangle + \epsilon_k)^2] & \langle \cdot, \cdot \rangle \text{ distributivity} \\
&= 2(\langle \theta_k - \theta^*, X_k \rangle + \epsilon_k) \nabla_{\theta_k} [\langle \theta_k - \theta^*, X_k \rangle + \epsilon_k] & \text{chain rule} \\
&= 2(\langle \theta_k - \theta^*, X_k \rangle + \epsilon_k) X_k \\
&= 2\langle \theta_k - \theta^*, X_k \rangle X_k + 2\epsilon_k X_k & (*)
\end{aligned}
$$

Taking expectation with respect to distribution $P_k$:

$$
\begin{aligned}
g_k(\theta_k) &= \mathbb{E}_{P_k}[\nabla_{\theta_k} L(Z; \theta_k)] && g \text{ def'n} \\
&= \mathbb{E}_{P_k}[2\langle \theta_k - \theta^*, X_k \rangle X_k + 2\epsilon_k X_k] && \text{by equation *} \\
&= 2\mathbb{E}_{P_k}[\langle \theta_k - \theta^*, X_k \rangle X_k] + 2\mathbb{E}_{P_k}[\epsilon_k]\mathbb{E}_{P_k}[X_k] && \mathbb{E} \text{ linear}, \epsilon_k \text{ indep.} \\
&= 2\mathbb{E}_{P_k}[\langle \theta_k - \theta^*, X_k \rangle X_k] && \mathbb{E}[\epsilon_k] = 0 \\
&= 2\mathbb{E}_{P_k}[X_k \langle \theta_k - \theta^*, X_k \rangle] && \text{scalar and vector commute} \\
&= 2\mathbb{E}_{P_k}[X_k \langle X_k, \theta_k - \theta^* \rangle] && \langle \cdot, \cdot \rangle \text{ commutative} \\
&= 2\mathbb{E}_{P_k}[X_k(X_k^T(\theta_k - \theta^*))] && \langle \cdot, \cdot \rangle \text{ matrix form} \\
&= 2\mathbb{E}_{P_k}[X_k X_k^T](\theta_k - \theta^*) && \text{matrix mult. associative} \\
&= B_k(\theta_k - \theta^*)
\end{aligned}
$$

In the last step, we have defined the matrix $B_k$ as twice the expected outer product of $X_k$ with itself, which is precisely the autocorrelation matrix, and is a function of $X_k$'s first two moments as follows.

$$
\begin{aligned}
B_k &= 2\mathbb{E}_{P_k}[X_k X_k^T] \\
&= 2\text{Corr}(X_k, X_k) \\
&= 2\left(\text{Cov}(X_k, X_k) + \mathbb{E}_{P_k}[X_k]\mathbb{E}_{P_k}[X_k]^T\right)
\end{aligned}
$$

So if there exists $k_1, k_2 \in \mathbb{N}$ such that the autocorrelation matrices differ

$$
\text{Corr}(X_{k_1}, X_{k_1}) \neq \text{Corr}(X_{k_2}, X_{k_2})
$$

then $B_{k_1} \neq B_{k_2}$. Assumption 1 provides the necessary $k_1, k_2$, so the proposition is proved.

$\square$

### A.4.3 ODE CORRESPONDENCE

In Section 3 of the main body, we defined $B(t)$, Lipschitz continuous in $t$, such that $B_k = B(\sqrt{\eta}k)$. If we assume that $\{X_k\}$ are sampled at intervals of $\sqrt{\eta}$ from an underlying continuous-time stochastic process $X(t)$ for which $\mathbb{E}_{P(t)}[X(t)]$ and $\text{Cov}(X(t), X(t))$ are Lipschitz continuous in $t$, this naturally induces a unique Lipschitz continuous $B(t)$. Note $P(t)$ is the distribution at time $t$, and we can extend equation 2 as follows.

$$
B(t) = 2\text{Cov}(X(t), X(t)) + 2\mathbb{E}_{P(t)}[X(t)]\mathbb{E}_{P(t)}[X(t)]^T \tag{8}
$$

This means that $\{B_k\}$ is sampled from a continuous $B(t)$, with subsequent $B_k$ spaced $\sqrt{\eta}$ apart. In essence, step-size $\eta$ acts as a scaling constant, such that tne unit of $t$ is $\approx \frac{1}{\sqrt{\eta}}$ discrete steps of $k$. (Exact when $\frac{1}{\sqrt{\eta}}$ is an integer.)

**Proposition 2.** *The SGDm iterates $\{\theta_k\}$ numerically integrate the ODE system equation 3 with integration step $\sqrt{\eta}$ and first order consistency.*

$$
\ddot{\theta}(t) + \frac{1 - \mu}{\sqrt{\eta}}\dot{\theta}(t) + B(t)(\theta(t) - \theta^*) = 0 \tag{3}
$$

*Proof.* The proof proceeds two parts. First we derive a first order operator-splitting integrator for the ODE equation 3, then we show that it is equivalent to SGDm equation 1.

*(Part 1: Operator-Splitting Integrator)*

Let $\xi : \mathbb{R} \mapsto \mathbb{R}^{2d}$ be the vector valued function of time whose first $d$ elements are $\theta(t)$, and last $d$ elements are $\dot{\theta}(t)$:

$$
\xi(t) = \begin{bmatrix} \theta(t) - \theta^* \\ \dot{\theta}(t) \end{bmatrix} \tag{9}
$$

i.e. $\xi$ is a phase space transformation allowing us to rewrite ODE equation 3 in the form of equation 4, which can then be split into a sum of separate systems $f^{[1]}, f^{[2]}$ as follows

$$\dot{\xi}(t) = f(\xi(t), t) \tag{10}$$

$$\dot{\xi}(t) = \begin{bmatrix} 0_{d \times d} & I_{d \times d} \\ -B(t) & -\frac{1-\mu}{\sqrt{\eta}} I_{d \times d} \end{bmatrix} \xi(t)$$

$$\dot{\xi}(t) = \begin{bmatrix} 0_{d \times d} & I_{d \times d} \\ 0_{d \times d} & 0_{d \times d} \end{bmatrix} \xi(t) + \begin{bmatrix} 0_{d \times d} & 0_{d \times d} \\ -B(t) & -\frac{1-\mu}{\sqrt{\eta}} I_{d \times d} \end{bmatrix} \xi(t)$$

$$\dot{\xi}(t) = f^{[1]}(\xi(t), t) + f^{[2]}(\xi(t), t)$$

Let $h > 0$ be an integration time step, $\phi_{h,k}^{[1]}(\xi_k)$ be the implicit Euler numerical flow of $f^{[1]}(\xi(t), t)$:

$$\phi_{h,k}^{[1]}(\xi_k) = \xi_k + h f^{[1]}(\phi_{h,k}^{[1]}(\xi_k), t_{k+1})$$

$$= \xi_k + h \begin{bmatrix} 0_{d \times d} & I_{d \times d} \\ 0_{d \times d} & 0_{d \times d} \end{bmatrix} \phi_{h,k}^{[1]}(\xi_k) \tag{11}$$

Let $\phi_{h,k}^{[2]}(\xi_k)$ be explicit Euler numerical flow of $f^{[2]}(\xi(t), t)$:

$$\phi_{h,k}^{[2]}(\xi_k) = \xi_k + h f^{[2]}(\xi_k, t_k)$$

$$= \xi_k + h \begin{bmatrix} 0_{d \times d} & 0_{d \times d} \\ -B(t_k) & -\frac{1-\mu}{\sqrt{\eta}} I_{d \times d} \end{bmatrix} \xi_k \tag{12}$$

The composed flow

$$\phi_{h,k} := \phi_{h,k}^{[1]} \circ \phi_{h,k}^{[2]} \tag{13}$$

is a sequentially split operator, which has splitting error order 1 because equation 10 is time-varying linear system (Faragó et al., 2011). The operators being composed, implicit and explicit Euler, are both order 1 consistent with their respective systems (Iserles, 2008). The overall order of consistency of a split operator is the minimum of splitting error, and the orders of the composed flows (Csomós & Faragó, 2008). So we have that equation 13 approximates equation 10 with order 1 consistency.

*(Part 2: SGDm Equivalency)*

We will now show that equation 13 is equivalent to SGDm when integration timestep $h = \sqrt{\eta}$, where $\eta$ is the SGDm step-size. We start with the definition of $\xi_{k+1}$ as the numerical flow of $\xi_k$:

$$\begin{aligned} \xi_{k+1} &:= \phi_{h,k}(\xi_k) \\ &= \phi_{h,k}^{[1]}(\phi_{h,k}^{[2]}(\xi_k)) && \text{by equation 13} \\ &= \phi_{h,k}^{[2]}(\xi_k) + h \begin{bmatrix} 0_{d \times d} & I_{d \times d} \\ 0_{d \times d} & 0_{d \times d} \end{bmatrix} \phi_{h,k}^{[1]}(\phi_{h,k}^{[2]}(\xi_k)) && \text{subst'n from equation 11} \\ &= \phi_{h,k}^{[2]}(\xi_k) + h \begin{bmatrix} 0_{d \times d} & I_{d \times d} \\ 0_{d \times d} & 0_{d \times d} \end{bmatrix} \xi_{k+1} && \text{def'n of } \xi_{k+1} \\ &= \phi_{h,k}^{[2]}(\xi_k) + h \begin{bmatrix} \dot{\theta}_{k+1} \\ 0_d \end{bmatrix} && \text{matrix mult.} \\ &= \left( \xi_k + h \begin{bmatrix} 0_{d \times d} & 0_{d \times d} \\ -B(t_k) & -\frac{1-\mu}{\sqrt{\eta}} I_{d \times d} \end{bmatrix} \xi_k \right) + h \begin{bmatrix} \dot{\theta}_{k+1} \\ 0_d \end{bmatrix} && \text{subst'n from equation 12} \\ &= \left( \xi_k + h \begin{bmatrix} 0_d \\ -B(t_k)(\theta_k - \theta^*) - \frac{1-\mu}{\sqrt{\eta}} \dot{\theta}_k \end{bmatrix} \right) + h \begin{bmatrix} \dot{\theta}_{k+1} \\ 0_d \end{bmatrix} && \text{matrix mult.} \\ \xi_{k+1} &= \xi_k + h \begin{bmatrix} \dot{\theta}_{k+1} \\ -B(t_k)(\theta_k - \theta^*) - \frac{1-\mu}{\sqrt{\eta}} \dot{\theta}_k \end{bmatrix} && \text{simplification} \end{aligned}$$

Recalling that $\xi_k = \begin{bmatrix} \theta_k - \theta^* \\ \dot{\theta}_k \end{bmatrix}$, the above immediately provides the following two recurrence relations:

$$\theta_{k+1} = \theta_k + h \dot{\theta}_{k+1} \qquad\qquad \dot{\theta}_{k+1} = \dot{\theta}_k + h \left( -B(t_k)(\theta_k - \theta^*) - \frac{1-\mu}{\sqrt{\eta}} \dot{\theta}_k \right)$$

Now let integration time step $h$ be the square root of SGDm step-size, $h = \sqrt{\eta}$, and define $v_k = \sqrt{\eta}\dot{\theta}_k$. Substituting $h, v_k$, we proceed via elementary algebra:

$$\theta_{k+1} = \theta_k + \sqrt{\eta}\left(\frac{v_{k+1}}{\sqrt{\eta}}\right) \qquad \left(\frac{v_{k+1}}{\sqrt{\eta}}\right) = \left(\frac{v_k}{\sqrt{\eta}}\right) + \sqrt{\eta}\left(-B(t_k)\theta_k - \frac{1-\mu}{\sqrt{\eta}}\left(\frac{v_k}{\sqrt{\eta}}\right)\right)$$

$$\theta_{k+1} = \theta_k + v_{k+1} \qquad\qquad v_{k+1} = v_k + \eta\left(-B(t_k)(\theta_k - \theta^*) - \frac{(1-\mu)v_k}{\eta}\right)$$

$$v_{k+1} = v_k - \eta B(t_k)(\theta_k - \theta^*) - (1-\mu)v_k$$

$$v_{k+1} = \mu v_k - \eta B(t_k)(\theta_k - \theta^*)$$

Note that $B(t_k) = B_k$, and that $B_k$ defines the time-varying gradients induced by covariate shift and linear regression to a linear target (Proposition 1). Hence, under those conditions, we have arrived at SGDm equation 1. The proof is complete.

$\square$

The reader may observe that, given a solution $\xi(t)$ to the system $\dot{\xi}(t) = A(t)\xi(t)$, the proof above shows that the SGDm iterates $\{\theta_k\}$ are precisely a first order numerical approximation of the first $d$ dimensions of $\xi(t) = \begin{bmatrix} \theta(t) - \theta^* \\ \dot{\theta}(t) \end{bmatrix}$, but the remaining $d$ dimensions are approximated only up to a scale factor $\sqrt{\eta}$ by iterates $\{v_k\}$. However, this is not an issue. Solutions $\theta(t)$ to the system equation 3 are embedded within the first $d$ dimensions of $\xi(t)$, so the remaining $d$ dimensions and iterates $\{v_k\}$ do not affect the result.

### A.4.4  Parametric Resonance for ODE Convergence and Divergence

**Theorem 1.** *When $B(t)$ is periodic such that $B(t) = B(t + T)$, the spectral radius $\rho$ of $\psi(T)$ characterizes the stability of solution trajectories of equation 3 as follows:*

- *$\rho > 1 \implies$ trivial solution $\theta(t) = \theta^*$ is unstable. All other solutions diverge as $\theta(t) \to \infty$ exponentially with rate $\rho$.*

- *$\rho < 1 \implies$ trivial solution is asymptotically stable, all other solutions converge as $\theta(t) \to \theta^*$ exponentially with rate $\rho$.*

*Proof.* The proof of this theorem relies heavily on the well-established mathematics of Floquet theory and the stability result is contained in Theorem 1.9 and Theorem 1.10 of (Halanay, 1966). Theorem 1.10 states that for a system of differential equations of the form

$$\dot{\xi} = A(t)\xi, \qquad A(t + T) = A(t) \tag{14}$$

where $A(t)$ is piecewise continuous, the stability of the trivial solution $\xi(t) \equiv 0$ is determined by the spectral radius of the system's *monodromy matrix* $M$ defined below

$$M = \psi^{-1}(0)\psi(T) \qquad\qquad \text{where } \dot{\psi}(t) = A(t)\psi(t)$$

*(Monodromy Matrix)*

Matrix-valued functions $\psi(t)$ are the system's *fundamental solution matrix*, and elementary existence results for linear systems allow one to choose a $\psi(t)$ such that $\psi(0) = I$ so that the monodromy matrix simplifies as

$$M = \psi^{-1}(0)\psi(T)$$
$$= I^{-1}\psi(T)$$
$$= I\psi(T)$$
$$M = \psi(T)$$

Below we denote the spectral radius of $\psi(T)$ (and hence of $M$) as $\rho$.

*(First Order Linear Form, Stability via Floquet)*

Below we show that equation 3 can be transformed to the form equation 14 with $A(t)$ continuous such that trivial solution stability of equation 14 is equivalent to stability of the solution $\theta(t) = \theta^*$.

Let $\xi(t)$ be the phase space transformation of $\theta(t)$, similarly to the proof of Proposition 2

$$\xi(t) := \begin{bmatrix} \theta(t) - \theta^* \\ \dot{\theta}(t) \end{bmatrix}$$

so that for each $t$, $\theta(t) \in \mathbb{R}^d$ and $\xi(t) \in \mathbb{R}^{2d}$. This means equation 3 (restated below)

$$\ddot{\theta}(t) + \frac{1 - \mu}{\sqrt{\eta}} \dot{\theta}(t) + B(t)(\theta(t) - \theta^*) = 0$$

is equivalent to the following first order linear form

$$\dot{\xi}(t) = A(t)\xi(t) \qquad \text{where } A(t) = \begin{bmatrix} 0_{d\times d} & I_{d\times d} \\ -B(t) & -\frac{1-\mu}{\sqrt{\eta}} I_{d\times d} \end{bmatrix}$$

Since $B(t)$ is periodic and continuous, which is stronger than the requisite piecewise continuity. Since all other submatrices of $A(t)$ are constant, we have that $A(t)$ is also periodic and (piecewise) continuous. Now Theorem 1.10 in (Halanay, 1966) immediately provides the following

- $\rho > 1 \implies$ trivial solution $\xi(t) = 0$ is unstable. All other solutions diverge as $\xi(t) \to \infty$ exponentially with rate $\rho$.

- $\rho < 1 \implies$ trivial solution is asymptotically stable, all other solutions converge as $\xi(t) \to 0$ exponentially with rate $\rho$.

Since $\xi(t) = \begin{bmatrix} \theta(t) - \theta^* \\ \dot{\theta}(t) \end{bmatrix}$:

- The trivial solution $\xi(t) = 0$ is equivalent to $\theta(t) = \theta^*$

- $\xi(t) \to 0$ is equivalent to $\theta(t) \to \theta^*$

- $\xi(t) \to \infty$ is equivalent to $\theta(t) \to \infty$

The theorem is proved.

$\square$

## A.5 Detailed descriptions of experiments

Below we provide details of the experiments from section 4 in the main body. The tables below contain details about the data generating process, learning algorithm and optimizer, and hyperparameter choices.

Experiments 4.1-4.4 introduce new data generating processes $\{X_k\}$ via their mean sequences $\{\bar{x}_k\}$. For those experiments, a sample trajectory is visualized for several values of the process' frequency tuning parameter, either $f$ or $T$. In addition to the sample trajectory, the frequency content of each process is shown in two ways: the power spectral density (PSD) of $\{X_k\}$ estimated via a long sample trajectory, as well as the PSD of the underlying mean sequence $\{\bar{x}_k\}$. As one would expect from the sampling relationship between $X_k$ and $\bar{x}_k$, the PSD of $\{X_k\}$ is simply the PSD of $\{\bar{x}_k\}$ with a noise floor.

Experiments 4.5, 4.6 reuse the process from experiment 4.4, so the visual depiction is elided.

### A.5.1 EXPERIMENT 4.1 DETAILS

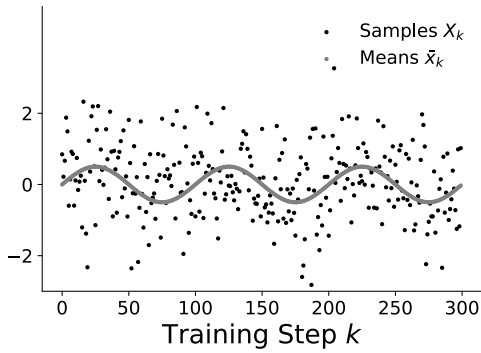

(a) Sinusoidal mean $\{X_k\}$ sample trajectory for $f = 0.01$

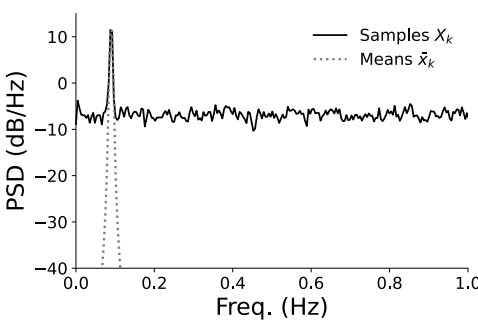

(b) Sinusoidal mean $\{X_k\}$ frequency content for $f = 0.01$

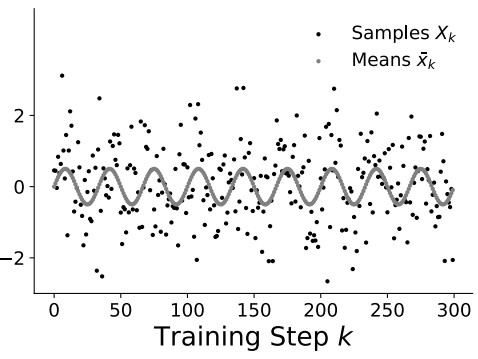

(c) Sinusoidal mean $\{X_k\}$ sample trajectory for $f = 0.03$

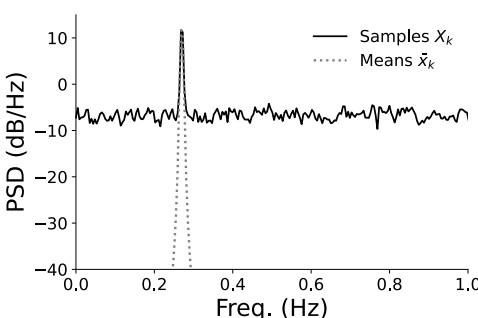

(d) Sinusoidal mean $\{X_k\}$ frequency content for $f = 0.03$

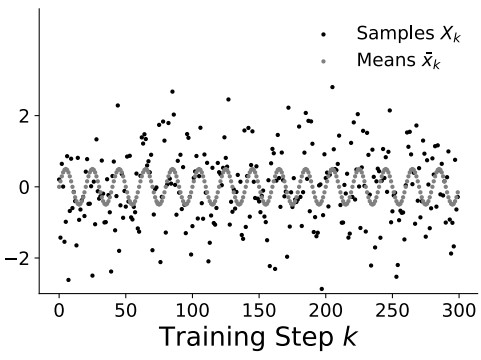

(e) Sinusoidal mean $\{X_k\}$ sample trajectory for $f = 0.05$

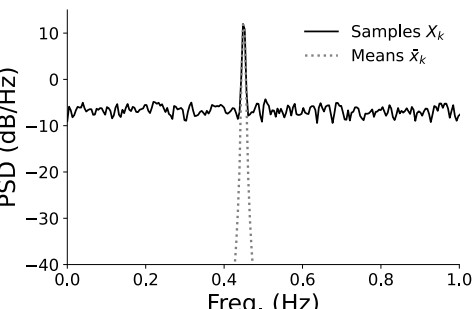

(f) Sinusoidal mean $\{X_k\}$ frequency content for $f = 0.05$

Figure 10: Sample trajectories (left) and power spectral densities (right) for Experiment 4.1 (Table 2) for three values of frequency tuning parameter $f$. The signal is a sinusoid, so we observe a single frequency peak translated left or right by $f$.

Table 2: Details for linear regression sinusoidal covariate shift problem.

| | |
|---|---|
| Sinusoid Mean Frequency | $f \in [0, 0.05]$ |
| Covariate Shift Mean | $\bar{x}_k = 0.5 \sin(2\pi f k)$ |
| Input Sampling | $X_k \sim \mathcal{N}(\bar{x}_k, 1)$ |
| Target Function (Fixed $\forall k$) | $Y_k = \theta_1^* X_k + \theta_2^*$ 
 $\theta_1^*, \theta_2^* \sim \text{Uniform}[-1, 1]$ |
| Model and Optimizer | $\widehat{Y_k} = \theta_{k,1} X_k + \theta_{k,2}$ 
 $\theta_{0,1}, \theta_{0,2} \sim \text{Uniform}[-1, 1]$ 
 $(\theta_k)_{k \in \mathbb{N}} \leftarrow \text{SGDm}(\eta, \mu)$ 
 $\eta = 0.01, \mu \in [0.95, 0.999]$ 
 $k \in [0, 10^4]$ |

### A.5.2 EXPERIMENT 4.2 DETAILS

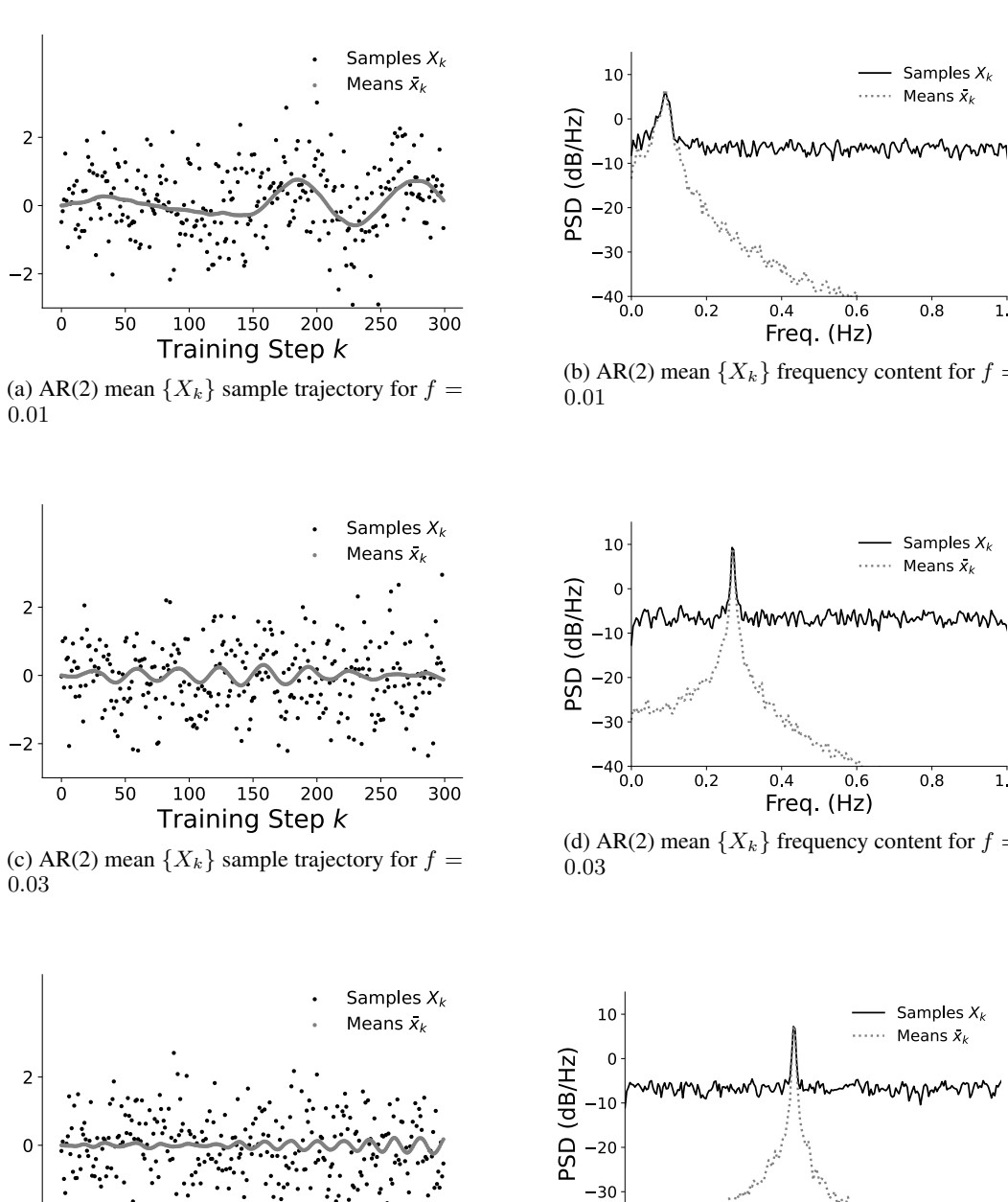

(a) AR(2) mean $\{X_k\}$ sample trajectory for $f = 0.01$

(b) AR(2) mean $\{X_k\}$ frequency content for $f = 0.01$

(c) AR(2) mean $\{X_k\}$ sample trajectory for $f = 0.03$

(d) AR(2) mean $\{X_k\}$ frequency content for $f = 0.03$

(e) AR(2) mean $\{X_k\}$ sample trajectory for $f = 0.05$

(f) AR(2) mean $\{X_k\}$ frequency content for $f = 0.05$

Figure 11: Sample trajectories (left) and power spectral densities (right) for Experiment 4.2 (Table 3) for three values of frequency tuning parameter $f$. The signal is a AR(2) process designed for a single frequency peak, as observed. The process is stochastic, so its frequency peak is widened by the imperfect correlation.

Table 3: Details for linear regression sinusoidal covariate shift problem. Rather than choosing a frequency $f$ and using it directly as in Table 2, we use $f$ together with the stationary distribution variance $0.1$ to compute AR(2) coefficients $\phi_1, \phi_2$.

| | |
|---|---|
| Expected Dominant Freq. in $\bar{x}_k$ | $f \in [0, 0.05]$ |
| $\bar{x}_k$ Stationary Dist. (Fixed $\forall f$) | $P = \mathcal{N}(0, 0.1)$ |
| Covariate Shift Mean | $\bar{x}_k = \phi_1 \bar{x}_{k-1} + \phi_2 \bar{x}_{k-2} + \xi_k$ 
 $\xi_k \sim \mathcal{N}(0, 10^{-5})$ iid 
 $\bar{x}_1, \bar{x}_2 \sim P$ 
 $\phi_1 = \dfrac{4\phi_2}{\phi_2 - 1} \cos(2\pi f)$ 
 $\phi_2$ s.t. $[\bar{x}_k \mid \bar{x}_{k-1} \sim P] \sim P$ |
| Remaining Parameters | $X_k, Y_k, \widehat{Y_k}, \theta^*, \theta_0, (\theta_k)_{k \in \mathbb{N}}, \eta, \mu, k$ same as Table 2 |

### A.5.3 EXPERIMENT 4.3 DETAILS

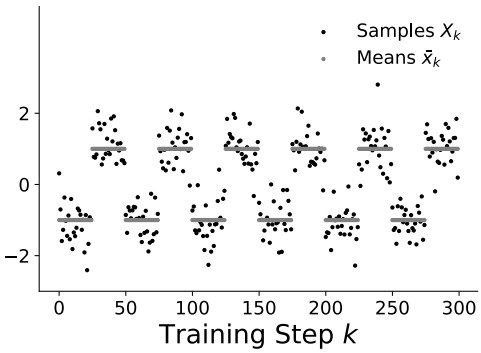
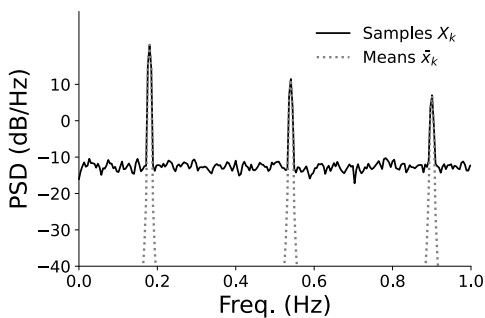

(a) Periodic mean $\{X_k\}$ sample trajectory for $T = 50$

(b) Periodic mean $\{X_k\}$ frequency content for $T = 50$

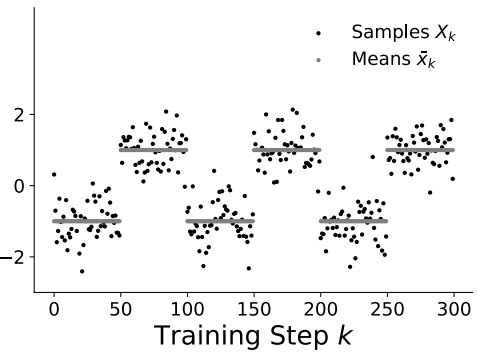
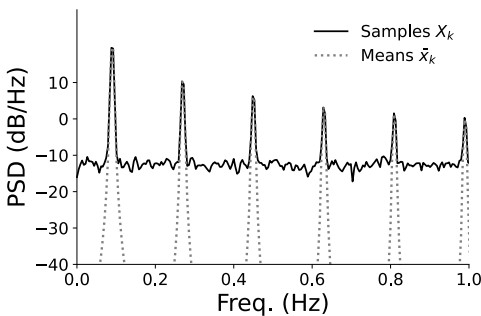

(c) Periodic mean $\{X_k\}$ sample trajectory for $T = 100$

(d) Periodic mean $\{X_k\}$ frequency content for $T = 100$

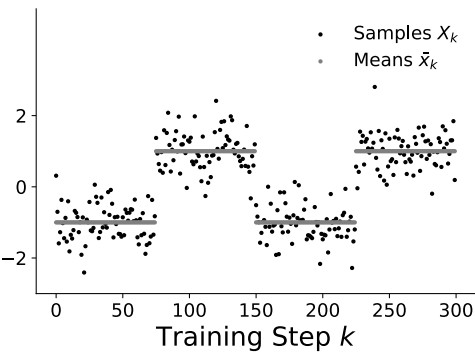
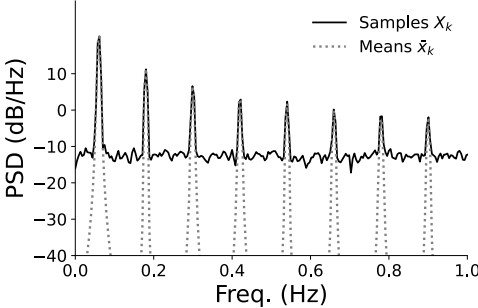

(e) Periodic mean $\{X_k\}$ sample trajectory for $T = 150$

(f) Periodic mean $\{X_k\}$ frequency content for $T = 150$

Figure 12: Sample trajectories (left) and power spectral densities (right) for Experiment 4.3 (Table 4) for three values of frequency tuning parameter $T$. The signal is a square wave in a single dimension for the figure, but is a square wave randomly oriented in higher dimensions for the actual experiment. There are multiple frequency peaks, with the frequency domain contracted by changing $T$.

Table 4: Details for linear regression square wave covariate shift problem.

| | |
|---|---|
| Square Wave Mean Period | $T \in [0, 120]$ |
| Input Dimensionality | $d = 5$ |
| Covariate Shift Mean | $\bar{x}_k = \begin{cases} \frac{\xi}{2\|\xi\|} & \text{if } \lfloor \frac{2k}{T} \rfloor \equiv 0 \mod 1 \\ -\frac{\xi}{2\|\xi\|} & \text{if } \lfloor \frac{2k}{T} \rfloor \equiv 1 \mod 1 \end{cases}$ 
 $\xi \sim \mathcal{N}(0, I_{d \times d})$ iid |
| Input Sampling | $X_k \sim \mathcal{N}(\bar{x}_k, 0.25 I_{d \times d})$ |
| Target Function (fixed for all $k$) | $Y_k = \langle \theta^*_{[1:d]}, X_k \rangle + \theta^*_{d+1} + \epsilon_k$ 
 $\theta^* \sim \mathcal{N}(0, c I_{d+1 \times d+1})$ where $c = 0.25$ 
 $\epsilon_k \sim \mathcal{N}(0, 0.1)$ |
| Model and Optimizer | $\widehat{Y_k} = \langle \theta_{[1:d]}, X_k \rangle + \theta_{d+1}$ 
 $\theta_0 \sim \mathcal{N}(0, c I_{d+1 \times d+1})$ where $c = 0.25$ 
 $(\theta_k)_{k \in \mathbb{N}} \leftarrow \text{SGDm}(\eta, \mu)$ 
 $\eta = 0.01, \mu = 0.95$ 
 $k \in [0, 10^4]$ |

### A.5.4 EXPERIMENT 4.4 DETAILS

Table 5: Details for linear regression problem with stochastically switching covariate shift mean.

| | |
|---|---|
| Mean Switching Interval | $T \in [0, 50]$ |
| Mean Switching Variance | $v = 0.25$ for Figure 4b, $v \in [0, 0.4]$ for Figure 4a |
| Input Dimensionality | $d \in [1, 9]$ for Figure 4b, $d = 5$ for Figure 4a |
| Covariate Shift Mean | $\bar{x}_k = \xi_i$ where $i = \left\lfloor \dfrac{k}{T} \right\rfloor$ 
 $\xi_i \sim \mathcal{N}(0, vI_{d \times d})$ iid |
| Remaining Parameters | $X_k, Y_k, \widehat{Y_k}, \theta^*, \theta_0, (\theta_k)_{k \in \mathbb{N}}, \eta, \mu, k$ same as Table 4 |

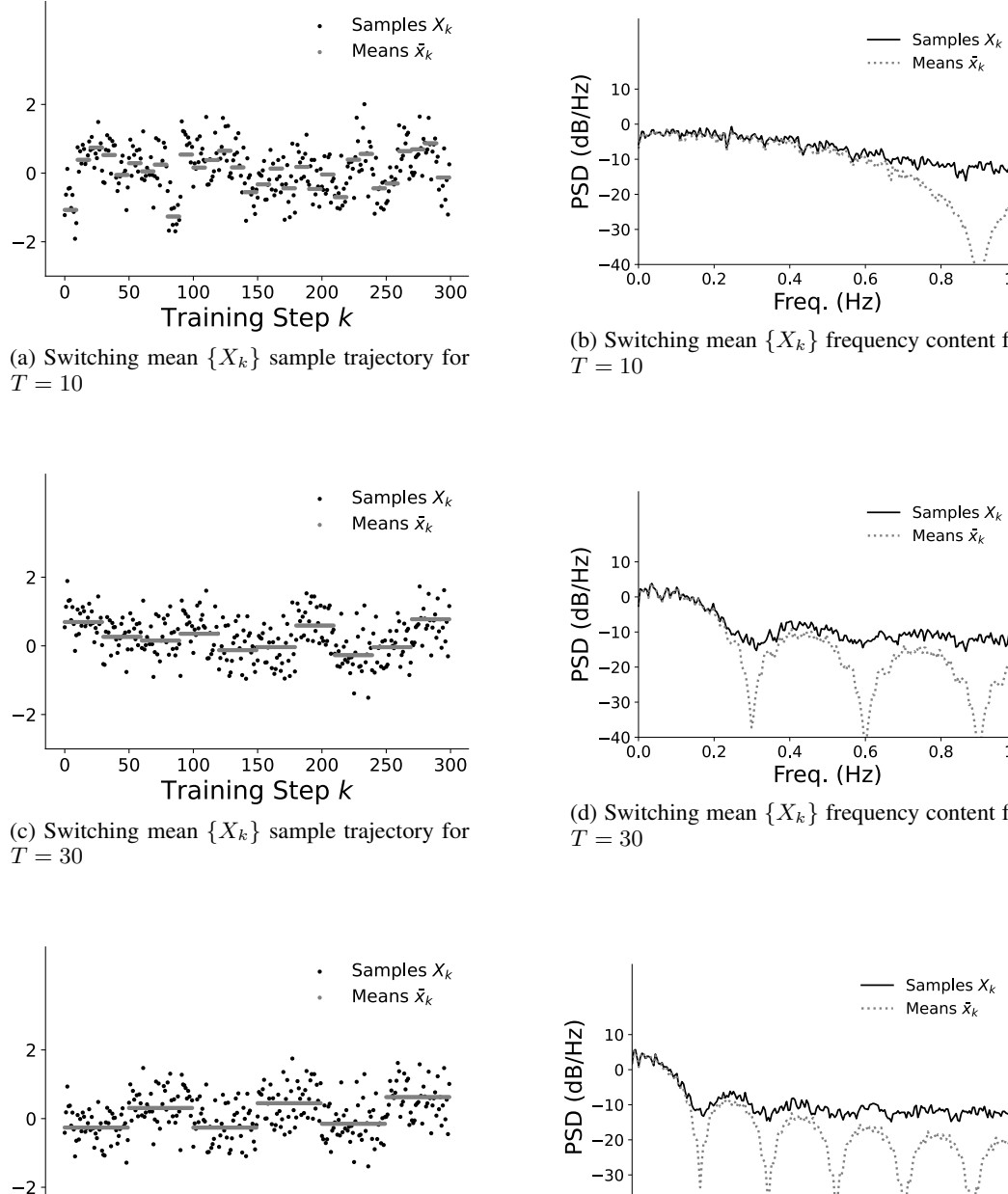

(a) Switching mean $\{X_k\}$ sample trajectory for $T = 10$

(b) Switching mean $\{X_k\}$ frequency content for $T = 10$

(c) Switching mean $\{X_k\}$ sample trajectory for $T = 30$

(d) Switching mean $\{X_k\}$ frequency content for $T = 30$

(e) Switching mean $\{X_k\}$ sample trajectory for $T = 50$

(f) Switching mean $\{X_k\}$ frequency content for $T = 50$

Figure 13: Sample trajectories (left) and power spectral densities (right) for Experiments 4.4-4.6 (Tables 4-7) for three values of frequency tuning parameter $T$. The mean signal is essentially white noise 'stretched' according to $T$. The signal is a single dimension for the figure, but is of higher dimensions for the actual experiment. The frequency content is a vertical reflection of Figure 12 (right,) with frequency troughs instead of peaks. Note that the frequency content is the mildest of all settings, having the lowest and least-defined peaks, which aligns with intuition: there is nothing even remotely oscillatory in its definition. It is reasonable to expect that many 'natural' processes would have more aggressive frequency content.

### A.5.5 EXPERIMENT 4.5 DETAILS

For visual depiction, see Figure 13.

Table 6: Details for linear regression problem with stochastically switching covariate shift, optimized with ADAM instead of SGDm.

| | |
|---|---|
| Mean Switching Interval | $T \in [0, 100]$ |
| Mean Switching Variance | $v = 1.0$ |
| Input Dimensionality | $d = 5$ |
| Covariate Shift Mean | $\bar{x}_k$ identical to Table 5 |
| Input Sampling | $X_k \sim \mathcal{N}(\bar{x}_k, 0.1I_{d \times d})$ |
| Target Function (fixed for all $k$) | $Y_k, \theta^*, \epsilon_k$ identical to Table 5 |
| Model | $\widehat{Y_k}, \theta_0$ identical to Table 5 |
| Optimizer | $(\theta_k)_{k \in \mathbb{N}} \leftarrow \text{ADAM}(\eta, \beta_1, \beta_2)$ 
 $\eta = 0.01, \beta_1 \in [0.9, 0.99], \beta_2 = 0.999$ 
 $k \in [0, 10^4]$ |

### A.5.6 EXPERIMENT 4.6 DETAILS

For visual depiction, see Figure 13.

Table 7: Details for neural network regression problem with stochastically switching covariate shift.

| | |
|---|---|
| Mean Switching Interval | $T \in [0, 100]$ |
| Mean Switching Variance | $v \in [0, 0.4]$ |
| Input Dimensionality | $d = 2$ |
| Covariate Shift Mean | $\bar{x}_k$ identical to Table 5 |
| Input Sampling | $X_k \sim \mathcal{N}(\bar{x}_k, 0.1I_{d \times d})$ |
| Target Function (fixed for all $k$) | $Y_k = \cos(\pi||X_k||) + \epsilon_k$ 
 $\epsilon_k \sim \mathcal{N}(0, 0.1)$ |
| Model | $\widehat{Y_k} = f(X_k; \theta_k)$ two hidden layers of 20 activations 
 $\theta_0$ initialized as He et. al. |
| Optimizer | $(\theta_k)_{k \in \mathbb{N}} \leftarrow \text{SGDm}(\eta, \mu)$ 
 $\eta = 0.01, \mu = 0.95$ 
 $k \in [0, 2 \times 10^4]$ |

