# OpenReview forum: "Resonance in Weight Space: Covariate Shift Can Drive Divergence of SGD with Momentum"
_ICLR.cc/2022/Conference — ICLR 2022 Poster_

### Official Review · Reviewer_Vijg · 2021-10-21

**Correctness:** 4
**Technical Novelty And Significance:** 4
**Empirical Novelty And Significance:** 3
**Recommendation:** 8
**Confidence:** 4

**Main Review:**

In general, I really like this study. It is very systematic and thorough, with theoretical considerations as a starting point, also very well written. I also like the comprehensive discussion of caveats and remaining construction sites. I believe the topic is important; iid sampling may be an idealization often difficult to achieve in practical settings.

Yet the potential practical implications is also one of the major critiques that I would have: Although I would agree that dependency is sometimes hard to avoid, I’m not sure how often it takes this clear-cut oscillatory form with one (or a few) dominant peaks in the power spectrum. So in my mind the consideration of at least one empirical dataset where this problem clearly arises is the most important omission from this paper. I would guess that in reality power spectra are much broader with a much wider range of frequencies covered. Does resonance, or at least a clearly suboptimal range, occur under these conditions?

My feeling is that also training algos like Adam are by now way more popular than SGDm, so it may be good to stress even more how and where the analyses give insights into the training process itself, and ways to potentially improve it even if resonance is not a huge issue. In general, more insight into why resonance problems do not arise with Adam would be helpful, or for which class of  optimizers we would expect them and which strategies protect against them.
Growing parameter oscillations that kick the algo out of local minima again are of course known for much longer, and not necessarily related to covariate shift but just too large learning rates or similar. Maybe it’s interesting to reflect on these connections here, as increasing \nu would be a major driver of instability here as well.

Some minor issues:
- Eq. 3 w/o time-dependent parameters would be a harmonic oscillator (for some param. settings). In addition it is assumed there are oscillations in the forcing input (in moments of X), which is the source of the resonance. This could be made a bit clearer; relates to the point about growing osc. above.
- Why is it necessary to assume in setting 4.2 that \phi_1 itself oscillates; the ARMA process already implements an oscillation?
- May be good to combine Fig. 3 & 4 just for layout.
- It’s shown (sect. 4.4) that the problem becomes more severe in higher dimensions. But couldn’t this be simply avoided by proper normalization of inputs, i.e. dividing by the whole vector norm prior to training?
- I missed whether 4.6 was performed with Adam or SGDm? I assume that even for high var. (0.4) the expected loss would go away for T → \infty?

**Summary Of The Paper:**

The paper discusses conditions under which SGD with momentum (SGDm) would diverge. Specifically, training samples are assumed to be non-iid, connected through oscillations in first or second moments. For the specific Gaussian setting of linear regression with oscillatory covariate shift, a driven (with time-dependent parameters) linear oscillator ODE is formulated for the parameters under SGDm. This allows to derive conditions, related to the learning and momentum rates, under which parameters would con-/diverge. Theoretical predictions are tested empirically, and it is shown that the basic phenomenon (resonance or at least suboptimal convergence) is also present in setups with noise, non-harmonic oscillations, other optimizers (Adam), and nonlinear regression models (NN).

**Summary Of The Review:**

Great and thorough paper, but empirical example that clearly demonstrates relevance would be appreciated.

---

> ### Author Response · Authors · 2021-11-18
> **Response to Vijg**
>
> Thank you for your thoughtful and detailed review, and for your positive feedback regarding the importance of the setting and the quality of the writing.
>
> We agree that experimentation using non-synthetic empirical datasets is an exciting and important future direction for our work, and that i.i.d. sampling is often difficult to achieve in practical settings.
>
> You offer a very reasonable critique in that real-world datasets might not display frequency peaks in the way that our synthetic data does, and we would like to offer a minor clarification: the synthetic data we train on was designed to have non-aggressive power spectra, with a single peak and a very high noise floor.  Our goal was to show that resonance is possible even with a minor step away from i.i.d. sampling.  After reading your feedback, we realize this was not clearly explained or justified in the paper.  If this paper is accepted, we will improve this in the camera ready version.  We believe that a short discussion and plots showing power spectra will be sufficient, but we are open if you have further suggestions!
>
> In light of that, your comment about empirical datasets is even more relevant, since there are types of natural data whose power spectra sharply peak around certain frequencies.  For example,  audio data, machine sensor data, weather data, traffic data.  Naturally oscillatory RL environments such as cart-pole, mountain car, and pendulum swing are also likely to have peaked power spectra in state observations.  We are just now beginning experiments with these problem settings and environments, and plan a comprehensive follow-up paper to focus on empirical and practical aspects, to build on the mathematical foundation of the current paper.
>
> It is true that algorithms such as Adam are now much more popular in use than SGDm, and we completely agree that resonance analysis with popular optimizers is interesting.  It bears mentioning that, even for SGDm’s simplest implementations, the phenomenon of acceleration in optimization is still not well understood, despite its more tractable analysis.  As written in Muehlebach, Jordan 2021:: “Yet, even for the class of strongly convex functions, most proofs that establish the superior convergence are algebraic and provide little qualitative understanding. As a consequence, there is little guide to the generality or robustness of the acceleration phenomenon across instances of optimization problems.”
>
> We’ve shown that SGDm under non-iid sampling is a parametric rather than a harmonic oscillator, and you are correct that oscillating hyperparameters (e.g. $\eta$) can be analyzed with the very same tools.  We appreciate this insight and exciting open problem.
>
> Minor Points:
>
> Regarding your point starting, “Eq. 3 w/o time-dependent parameters...“ could you clarify what you mean?
>
> In setting 4.2, the parameter $\phi_1$ does not actually oscillate.  The frequency $f$ is fixed per training run.  Sorry that was unclear, we will clarify this in the paper.
>
> Yes, it is certainly possible that the relationship between resonance and increasing input dimensionality could be due to increasing norm!  We will run the same experiments with normalized inputs and include them in the paper.
>
> 4.6 was performed with SGDm, and you are correct that the loss decreases as $T \to \infty$.  Indeed, this is one way to approach the i.i.d. data setting!

---

### Official Review · Reviewer_Vbeb · 2021-10-26

**Correctness:** 4
**Technical Novelty And Significance:** 2
**Empirical Novelty And Significance:** 3
**Recommendation:** 6
**Confidence:** 4

**Main Review:**

This paper points out the influence of data distribution pattern to the performance of optimization algorithms when the data are not sampled iid. It is important in the fields of reinforcement learning, online learning, etc. Both theoretical and numerical investigation are made, providing strong evidence for the existence and universality of the problem. Yet, there may be space for improvement, especially in the theoretical part. The reviewer has some major comments listed below:

1. The authors mentioned that they cannot show the convergence of SGDm when the continuous dynamics converges, because as a numerical discretization of the continuous ODE the numerical error accumulates and cannot be controlled. There is no problem with the argument provided by the authors. However, it is still possible to prove the convergence of the discrete dynamics (the SGDm), because the dynamics is linear. Here, instead of looking at the eigenvalue of the coefficient matrix of the continuous dynamics, we can directly look at that of the discrete dynamics. Then, we may be able to characterize the convergence of SGDm, which makes the picture more complete. Hence, the reviewer suggests the authors to do this analysis.

2. The "Example" part from page 4 to page 5 provides little information, because all the results and derivations are put to the appendix, and only descriptive language is used. If the authors feel the results unimportant, then you can just delete this part. But I don't think it unimportant, because this is the quantitative description of resonance phenomenon that lies on the focus of this paper. So please add more concrete results here. For example, the values of \eta, \mu, and f that result in resonance. On the other hand, section 5 is too long and can be made shorter.

3. The numerical results cover much more general cases than the theory, and show a general connection between performance of SGDm and data covariance shift. However, it is hard to say this connection, as shown by figures for nonlinear models like neural networks, originates from the same mechanism as the resonance. Is SGDm still shows an oscillating behavior in this case? Could the authors provide more evidence that the experimental results for neural networks are also caused by a match in oscillation frequencies? Are there other reasons for the change of SGDm performance?

**Summary Of The Paper:**

This paper studies the behavior of SGD with momentum when the training data are not iid. The authors consider a linear regression problem with covariance drift. In this case, they show that SGDm is the Euler's method for a second order ODE with a changing coefficient. Under the assumption of periodic covariance of the data, the authors identify a resonance phenomenon between the oscillation of data covariance and the oscillation of SGDm iterator. SGDm is proven to diverge when resonance happens. Though, when there is no resonance the authors are unable to prove the convergence of SGDm.

Beside the theory, numerical experiments are conducted for cases covered and not covered by the theory. Numerical results show a dependence of the optimizer performance and the covariance pattern in much broader settings than linear regression, even including neural networks.

**Summary Of The Review:**

The paper studies the influence of data covariance shift to the performance of SGDm. It is important to some machine learning applications. The authors uncover a resonance effect as a mechanism behind this influence. The insight is novel, the numerical experiments are extensive, while the theoretical study can be improved.

---

> ### Author Response · Authors · 2021-11-18
> **Response to Vbeb**
>
> Thank you for your thoughtful and detailed review.
>
> Based on your summary, it you have understood the paper well. I think it is likely just a difference in terminology, but we would like to clarify that the non-iid sampling we focus on is covariate shift, not covariance shift. When we say covariate shift we simply mean that samples of the inputs (covariates) {X_k} are non-iid as per Assumption 1. This is more general than the covariance shifting over time, although covariance shift is one way to achieve it.
>
> Responses to main points:
>
> 1. Analysis of discretizations of time-varying systems such as we consider can be exceptionally challenging. We focus on the continuous time dynamics and gradient flow because it allows us to leverage theoretical tools that are not readily available in the discrete setting. We agree that it is slightly unsatisfying that we are not able to prove convergence of SGDm under non-iid sampling due to the error rates, but it is as yet unclear how this should be done in the discrete case. Convergence for SGDm has been recently shown for Markovian sampling in Doan et al., 2020 (currently in preprint only), but the assumptions are quite different than those we consider.
>
>
> 2. We appreciate your feedback on this point. The example was longer in a previous iteration of the paper and was shortened for space reasons, but upon review after reading this comment we agree that it is no longer useful. We will move the example to the appendix in order to include a section to provide more intuition for the theoretical results. We have included a summary at the end of Section 3, and any feedback on its content would be highly appreciated!
>
>
> 3. This is a good point and nice observation. It is true that when we move into the non-linear case we lose our connection with the ODEs we present in the theory section, and it is thus reasonable to ask whether or not the divergence of SGDm we observe is in fact due to resonance. That being said, we are fairly confident that the performance we observe is in fact due to resonance. The machine learning task is extremely simple - E[y|x] is a fixed smooth function - and the neural networks easily learn this under iid sampling, as well as non-iid sampling that does not result in resonance in our linear models (e.g. non-iid sampling with vey uniform power spectral densities). We originally ran experiments with perfectly periodic covariate shift and non-linear models and observed worse performance of the models (i.e. worse convergence of model parameters) than we observed with the experiments included in the paper. All this suggests to us quite convincingly that it is indeed resonance we are observing, even though we lack the theoretical tools to precisely describe it. Resonance is somewhat dampened in the non-linear models as compared to the linear models, but they follow the same patterns in terms of the type of covariate shift that result in more/less resonance.
>
> We notice that you gave a score of only 2 for the technical novelty and significance. We believe that the technical aspect work is in fact very novel and are slightly confused by this low score. Our work is (to our knowledge) the first investigation of frequency response or resonance in any optimizer.  Such input-response analyses require a system and an input signal from which the system may be driven.  To our knowledge, covariate shift has never been proposed as a potential driving signal to the learning system, and our work is the first to propose and formalize this idea.  The result--that SGDm under covariate shift is a parametric oscillator--is of independent significance, because the wealth of existing literature on parametric oscillators (e.g. Chechurin and Chechurin, 2018) may now be readily applied.  Resonance implying divergence is simply an example.
>
> We took several other theoretical routes to attempt to explain the phenomena we observed, but eventually found mathematical machinery that elegantly described the divergence we observed. The existing results we depend upon are from quite disparate fields, and this paper links those fields in ways that are entirely novel, and valuable to the ML community.  That they appear straightforward and simple is actually the product of a great deal of work!
>
> If you feel we have addressed some of your concerns with the paper, we would appreciate you considering raising your score.  And we are happy to engage in further discussion!

---

### Official Review · Reviewer_Z17i · 2021-11-02

**Correctness:** 4
**Technical Novelty And Significance:** 4
**Empirical Novelty And Significance:** 3
**Recommendation:** 8
**Confidence:** 2

**Main Review:**

I found the paper to be quite well written and exemplary in its scientific format. A hypothesis is proposed about a phenomenon, the relevant math is derived, and then tested empirically in various settings that help us assess the correctness of the hypothesis and the relevance of the concept when departing from restrictive assumptions about the setting.

I think a weakness of the paper is that more could have been done to display the oscillation behaviours of deep neural networks. Some analyses that hold up to larger problems and deeper networks which may detect oscillation include: gradient interference, mode connectivity, or simple parameter projections. That being said I do think the current empirical content of the paper is valuable.

I'm unfortunately unable to assess the novelty of this paper. In terms of impact, coming from deep RL, these feel like very valuable insights which might actually inform my research; it's still possible that these are new insights to me only because I'm not familiar with ODE and oscillator analogies of optimizers as they exist in the supervised learning literature.

**Summary Of The Paper:**

This paper analyses the phenomenon of resonance in momentum SGD (SGDm) under a time-dependent covariate shift. This setting is useful to depart from iid-sampling-based theory, and could have useful implications for continual learning and reinforcement learning.
In the paper, it is shown that such a setting corresponds to a _parametric_ oscillator of the parameters, which explains instability and divergence in a non-iid SGDm setting.
The authors then test their hypothesis empirically, first in a simple setting that respects all the assumptions of the theorem, and then progressively getting rid of more and more assumptions, which ends up in finding a similar (albeit heavily dampened) phenomenon in non-linear ReLU-MLPs using Adam.


**Summary Of The Review:**

I recommend accepting this paper. The empirical results which I am able to evaluate are correct, useful and informative. I cannot attest to the mathematical validity of the paper, unfortunately.

---

> ### Author Response · Authors · 2021-11-18
> **Response to Z17i**
>
> Thank you for the thoughtful review and positive feedback. Based on your summary of the paper, it appears that you understood it well, despite not having much background with gradient flow methods for analyzing optimization algorithms  in supervised learning as you mention.
>
> We agree that further experimentation with deep neural networks is an exciting future direction for this work. Also, perhaps of particular interest to you coming from deep RL, we are currently in the process of running experiments on naturally oscillatory RL environments such as cart-pole, mountain car, and pendulum swing to examine the existence of resonance phenomena in these environments. As this work is the first of its kind in terms of analyzing non-iid sampling through the use of a time-varying system of ordinary differential equations, we wanted to focus on the linear setting and carefully controlled experiments in which we could progressively relax theoretical assumptions in order to build a solid foundation for this mode of analysis going forward. We believe that the perspective we develop in this paper offers the promise of further insight into the behaviour of SGDm under non-iid sampling and intend to pursue several directions to extend this paper in future work. One of these directions will certainly be more detailed experimentation with deep neural networks.

---

> > ### Comment · Reviewer_Z17i · 2021-11-29
> > **Re**
> >
> > Thanks for your response. I will try to take all of this into account in the upcoming discussion.

---

> > > ### Author Response · Authors · 2021-11-29
> > > **Re Z17i**
> > >
> > > We appreciate your willingness to discuss.
> > >
> > > A small tangent: we also appreciate your comments regarding our work's "exemplary scientific format."  From the very beginning of this project, we strove towards the _phenomenon --> hypothesis --> analysis_ structure you described.  So it is helpful feedback (and rewarding) to see that the structure remains clear!

---

### Official Review · Reviewer_6uKo · 2021-11-04

**Correctness:** 3
**Technical Novelty And Significance:** 2
**Empirical Novelty And Significance:** Not applicable
**Recommendation:** 3
**Confidence:** 3

**Main Review:**

Strengths, weaknesses, and comments

1. I think the paper tackles an important question, namely the convergence of optimization methods under non-iid samples. I think the phenomenon that the paper identifies, i.e., divergence of SGD with momentum due to certain resonant frequencies in covariate shift, is an interesting observation.

2. However, my overall assessment of this paper is that the contributions are either limited or poorly presented. Proposition 1 is a result of rather straightforward and elementary calculations. The proof of Proposition 2 is very similar to Section 2.3 of (Muehlebach and Jordan, 2021) because they also use a forward Euler update of the momentum coordinates, and then use the newly computed momentum for the position update. The key difference that the ODE is time-varying is taken care of by existing results in numerical analysis (i.e., Faragó et al., 2011, as cited in the paper), so the paper's technical contribution looks somewhat weak to me. As mentioned in the paper, Theorem 1 also "relies heavily on the well-established mathematics of Floquet theory." To me it looks like a direct application of the results in (Halanay, 1966).

3. Moreover, I believe that the presentation and discussion of Theorem 1 have big room for improvement. Theorem 1 is stated in terms of the fundamental solution matrix $\psi(t)$ and its spectral radius, and it is impossible to see how different quantities in the algorithm/data (e.g., $\mu$, $B(t)$, $T$) are related to them. When I read the paper for the first time, it wasn't obvious to me at all how Theorem 1 connects to the "resonance" behavior and why only some specific frequencies lead to divergence. Even the "Example" paragraph does not explain this resonance behavior, and it is left unclear why one gets the contours in Figure 2a. I think the solutions $\psi(t)$ are obtained numerically and it is hard to get any closed form solutions, but at least providing a more detailed explanation on the frequency responses and "peaks" should be helpful.

4. Also, I question if this paper is analyzing the "right" algorithm. The paper claims to analyze SGD with momentum, but the theoretical results only concern the version where *expected gradients* are used for the updates. By taking expectation over the joint distribution of $X_k$ and $Y_k$ at every step, the stochasticity in the data samples is essentially removed. I'm in doubt if this can really be considered "SGD"?

5. From Figure 2(a), it seems like choosing $\mu \leq 0.96$ can avoid the resonance. Also, Section 4.3 shows that the resonance is dampened when we sample less points. While the divergence phenomenon is interesting, these two facts make me think that the phenomenon might only occur in very limited settings (i.e., $\mu$ close to 1 and small noise in stochastic gradients) and hence not very relevant to practical situations.

6. Although Theorem 1 characterizes convergence and divergence of the continuous-time ODE, the discussion after Theorem 1 mentions that only the divergence part of Theorem 1 has implications for discrete-time SGDm. If our goal is to show only divergence in SGDm, I'm curious: wouldn't it be easier to analyze the (discrete-time) SGDm directly and show its divergence, rather than relying on its corresponding ODE?

Minor comments/questions

7. I believe there is a mismatch between Section 2 problem setting versus what is actually studied and tested. In Section 2 it is mentioned that the covariate $X_k$'s marginal distribution converges to some stationary distribution $\Pi$ over time. However, the ones studied later do not follow this setting. For example, the periodic covariate shift in Section 4.1 does not converge to a stationary distribution.

8. The sentence in the beginning of Section 3, "We show the parametric resonance conditions necessary to induce exponential divergence in SGDm," reads as if parametric resonance is a *necessary* condition for divergence in SGDm. However, I believe this is not the case; the paper is providing a sufficient condition for divergence.

9. In Section 3.1, "regression with covariate shift and a stationary target distribution": why is target distribution stationary when the distribution of $X_k$ changes? Maybe the authors meant that the coefficient $\theta^*$ is fixed, and the distribution of $\epsilon_k$ is stationary?

10. In Assumption 1, can't there be a situation where the expectation of $X_{k_1}$ and $X_{k_2}$ are different and the covariance matrices also differ, but they both have the same matrix $B_{k_1} = B_{k_2}$ (eq. (2))?

11. Proof sketch of Theorem 1: what is $g(t, \theta(t))$?

12. Pages 20 & 21: In the recurrence relations for $\dot \theta$ and $v$, $\theta_k$'s should instead be $\theta_k - \theta^*$?


**Summary Of The Paper:**

This paper studies the convergence of SGD with momentum under linear regression with covariate shift. Data comes from $Y_i = <\theta^*, X_i> + \epsilon_i$ for a fixed $\theta^*$ and iid zero-mean noise $\epsilon_i$, but the distribution of $X_i$ varies over time ("covariate shift"), leading to non-iid samples. In this setting, the paper shows that the expected progress of SGD with momentum is a discretization of a second-order ODE. Then, the paper proceeds to show that if the covariate shift is periodic, the convergence/divergence of the ODE can be determined by looking at the spectral radius of a matrix called the monodromy matrix. The paper moves on to show experiments that test the theoretical characterizations and also investigate if different settings under relaxed assumptions show similar "resonance" behavior.

The keyword "resonance-driven divergence" in this paper can be understood as follows: divergence of SGD with momentum happens only at some specific frequency of covariate shift. Metaphorically, this is similar to breaking a wine glass with a sound wave tuned to the right frequency.


**Summary Of The Review:**

Although the paper studies an interesting setting and identifies an interesting failure mode of momentum methods, I believe the theoretical contributions as well as implications to practice are not strong enough to grant acceptance (comments 2 & 5). There is room for improvement in terms of presentation (comment 3), and also the analysis is carried out only using the expected gradients which I find somewhat doubtful (comment 4).

---

> ### Author Response · Authors · 2021-11-18
> **Response to 6uKo**
>
> Thank you for reading and understanding the paper so thoroughly. Your feedback has been extremely helpful. Points 8, 9, 11, and 12 were simple writing mistakes on our part, and the paper has been updated accordingly. We address the remaining points below.
>
> **Points 2. and 3.** Upon reading your feedback we agree that there are issues with theory presentation, and that the example in Section 3 is not helpful in clarifying it. In the revised submission, we will move the example paragraph to the appendix where the rest of the example already sits, and replace it with a summary of the main mathematical objects and how they affect each other. The new content follows, any feedback is highly appreciated:
>
> > **Summary:** There is a chain of dependencies starting from the non-i.i.d. sampling in $\{X_k\}$, and ending at the monodromy’s spectral radius.
>
> > $\{X_k\} \to X(t) \to B(t) \to A(t) \to \psi(t) \to \psi(T) \to \rho$
>
> > In order, we have the discrete stochastic process $\{X_k\}$ from which training inputs are sampled, and its underlying continuous stochastic process $X(t)$.  If $\{X_k\}$ and $X(t)$ were i.i.d., then the matrix $B$ would be constant, but the non-i.i.d. nature means $B(t)$ is a function of time.  The matrix $A(t)$ is the matrix describing SGDm’s continuous time dynamics as a linear time-varying ODE $\dot{\xi} = A(t) \xi$, and the matrix $B(t)$ is simply a submatrix of $A(t)$.  Since we have a linear ODE, the space of solution trajectories is spanned by the columns of the ODE’s fundamental solution matrix $\psi(t)$.  Moreover, since we have a periodic $A(t)$ with period $T$, the stability of all solutions can be determined simply from the largest eigenvalue of $\psi(T)$, a.k.a. its spectral radius $\rho$.
>
> > The intuition behind $\rho$’s importance comes from the following fact: after each elapsed period $T$, the phase space (including weight space) is subject to the linear transformation $\psi(T)$.  So as time increases, say $n$ periods, the linear transformation is applied iteratively as $\psi(T)^n$, so it is clear that any eigenvalue larger than unity means all solutions eventually diverge exponentially.  This is precisely the parametric resonance condition.
>
> Regarding your points about limited and/or weak technical contributions, we agree that the mathematical machinery is simple, but we see this as fortunate, not disadvantageous. We would like to gently emphasize that publishability should not be determined by the challenge posed to readers by its mathematics. We are confident that our theory is correct and supports our claims, which is the relevant part. We attempted several alternative formalisms and techniques, some of which were highly complex. In the end, we discarded all others in favour of numerical integration and Floquet theory, because its predictions align with observed phenomena extremely accurately.
>
> Regarding originality and significance, our work is (to our knowledge) the first investigation of frequency response or resonance in any optimizer.  Such input-response analyses require a system and an input signal from which the system may be driven. To our knowledge, covariate shift has never been proposed as a potential driving signal to the learning system, and our work is the first to propose and formalize this idea. The result--that SGDm under covariate shift is a parametric oscillator--is of independent significance, because the wealth of existing literature on parametric oscillators (e.g. Chechurin and Chechurin, 2018) may now be readily applied. Resonance implying divergence is simply an example.
>
> **Points 4, 5, 6.** These are excellent questions and observations. They are relevant to more works than just ours--both extant and future--and we are happy to discuss here. Since responses are limited to 5000 characters, we defer discussion to a separate comment.
>
> **Point 7.** You’re correct that we don’t need there to be a stationary distribution in order for resonance to occur. We invoked the stationary distribution in order to define the minimization problem which SGDm is solving, and to directly connect our work to existing literature.
>
> However, the periodic covariate shift of Experiment 4.1 indeed induces a stationary distribution, even though the distribution of $\{X_k\}$ does not converge as $k \to \infty$ grows. We can provide an illustrative example if desired!
>
> **Point 10.** You are correct. After further consideration, the quantity of interest is actually the autocorrelation matrix $Corr(X, X) = E[X X^T]$.  Thankfully, this is a trivial change to the paper, as the autocorrelation matrix is the quantity from which equation (2) is derived. We will modify Assumption 1 to be “...we have inequality in autocorrelation matrix $Corr(X_{k_1}, X_{k_1}) \neq Corr(X_{k_2}, X_{k_2})$.”  The final four lines of Proposition 1’s proof will change accordingly.
>
> Additional Citation
>
> Chechurin, S, and Chechurin,L. *Physical Fundamentals of Oscillations.* Germany, Springer, 2018.

---

> > ### Comment · Reviewer_6uKo · 2021-11-29
> > **Response acknowledged---sorry for last-minute comment**
> >
> > Dear authors,
> >
> > I appreciate the authors for their detailed response, which addressed and clarified many of my concerns about this paper. However, I believe that some central points in my concerns are not resolved properly. I know this is a late moment to post such a critical post that could use the authors' response, but I thought that it is better late than never. Below, allow me to share my thoughts on the authors' response to points 2-4.
> >
> > 2. I totally agree with the authors that simplicity in math is a bliss, but I was not criticizing the paper just based on the simplicity. My point was on the *novelty* of the mathematical techniques presented in the paper. I love simple and elegant proofs, but when it comes to papers I review, I believe that there has to be some novelty in it. My criticism was on the fact that the paper claims "technical novelty" (the end of Section 3) while handing over the "dirty job" to other existing papers. For example, regarding the difficulty of dealing with time-varying ODE, it is claimed at the end of Section 3 that "Our proof specifically addresses this difficulty using operator splitting theory from recent numerical integration literature." However, from Section A.3.3 of this paper, the proof is not giving any special treatment to the time-varying nature of $B(t)$, but relegating it to existing results such as (Faragó et al., 2011, etc.). While the argument looks technically correct and the authors indeed deal with time-varying $B(t)$, I find the claimed "technical novelty" rather questionable.
> >
> > 3. While the added "chain" clarifies the dependencies to some extent, I feel that the discussion still does not provide good intuitions on when/why resonance happens. For example, for the mean sequence $0.5 \sin (2\pi f k)$ tested in experiment 4.1, why do we observe resonance specifically around $T = 45$ and $22.5$? If the authors can provide a more intuitive explanation than just "because $\rho > 1$ for such $T$'s", then I think the paper can deliver useful insights.
> >
> > 4. I understand that the experiments, even with stochasticity, may align quite well with the theoretical predictions. However, it is mentioned in Section 4.3 that the resonance is dampened as we sample less and less points, i.e., with more and more stochasticity. If the "S" in SGDm makes the theoretical results less aligned with experiments, is it fair to claim anything about "S"GDm based on the theory developed in this paper? The paper makes two levels of abstraction in the analysis of SGDm. The first is taking out the "S", and the other is considering the continuous-time variant. For SGD without momentum, this is equivalent to making claims about SGD based on some theoretical results on gradient flow. I believe this can be highly misleading.
> >
> > Again, I am sorry for posting my comment at the last minute. Since I am the only negative reviewer, I intentionally tried to be more critical. I hope this comment clarifies my points better.

---

> > > ### Author Response · Authors · 2021-11-29
> > > **Extended discussion with reviewer 6uKo**
> > >
> > > We thank the reviewer for reading our response, and for further clarification!  Your additional details are clarifying on all points.  We are excited to respond.
> > >
> > > 2. **Novelty.**
> > >
> > >    We agree and would like to soften language around what is specifically novel.  We definitely do not want to over-claim novelty which is not ours.
> > >
> > >    We see your point about novelty proof technique, and we agree that our language should be softened.  This is indeed a paper connecting two disparate fields: nonlinear resonance theory and machine learning.  We claim novelty because it is a novel connection between the two, but it is not novel in proof techniques.  The main novelty of the paper the observation and explanation of nonlinear resonance phenomenon occurring in an ML context.  Would the reviewer's concern on this point be assuaged by removing claims of proof technique novelty?
> > >
> > > 3. **Ball and bowl intuition.**
> > >
> > >    We again agree.  We offer some further intuition below based on a physical analogy.  We would appreciate feedback on whether or not it is helpful!  If yes, we are happy to include it in the paper.
> > >
> > >    The summary description added alongside the dependency chain is truly the most intuitive explanation we can muster while staying formal.  However, if formality is relaxed, there is a nice intuition, where the loss surface is likened to a quadratically-shaped bowl warping over time, with the weights propagating through the bowl as a ball with momentum.  This is an imperfect analogy, because a ball has angular momentum, and a quadratic bowl does not actually induce a linear potential well.  But the time-varying intuition is valuable.  We have a large figure which graphically depicts this, but are unaware of a way to share the figure here.  A textual description follows.
> > >
> > >    In the i.i.d. setting, the expected loss surface does not move, so the bowl is fixed in space and the ball will descend towards the global minimum (possibly with some oscillation at the system's natural frequency) as friction exponentially depletes the ball's velocity.  The covariate shift described in the paper results in the bowl's surface warping in an oscillatory folding motion.  While the fixed point of the bowl does not move, the sides of the bowl do, and they can push the ball along its path.  If the bowl's folding motion occurs at the system's natural frequency (or better yet, twice the natural frequency), then the ball will gain energy from the bowl, and the trajectory will exponentially escape.  The fact that the strongest resonance occurs at twice the natural frequency, and the fact that escape is exponential, is due to the effect's nonlinearity.  Linear resonance occurs at simple harmonics, and trajectory escape requires zero friction, achieving only linear growth.
> > >
> > > 4. **S in SGD.**
> > >
> > >    Upon reviewing the figure for experiment 4.3, we agree that it does not support the claim that ignoring stochasticity was theoretically appropriate.  We do believe that it was the correct choice, and we hope to convince the reviewer and future paper readers with other experimental results.  Those results are complete, and we have described them below.
> > >
> > >    The results of experiment 4.3 makes the damping effect induced by stochasticity appear very strong, and that a reader would be justified in questioning whether or not there is a fundamental difference in the fully stochastic setting.   We should not rely on that as evidence that the stochasticity does not affect theoretical validity.  We have repeated experiments 4.1 and 4.2 with a single sample per step, instead of 20 per step.  This represents a much greater step away from expected gradient than 4.3, and the experimental results are qualitatively identical, with the same two 'bulbs' of divergence in the same region.  The heatmap is simply more noisy and slightly attenuated.  This does not conflict with the results of 4.3, because its drastic difference in curve height is irrelevant, because even small trajectory growth will eventually grow arbitrarily, and because the frequency content differences along its x-axis are not especially large.
> > >
> > > Apologies for any sloppiness in our reply.  We wanted to continue this discussion before time ran out, as your feedback has been highly valuable.

---

> ### Author Response · Authors · 2021-11-18
> **Response to 6uKo, Additional Discussion**
>
> Since comments are limited to 5000 characters, we separately address points 4-6 here. We again thank the reviewer for such thorough feedback.
>
> **Point 4.**  This is an excellent question regarding the relevance of expected-gradient-based analysis to stochastic descent algorithms. We care about the S in SGDm because we care about the online setting, where we have only a sample from each $X_k$, and we do not have access to the distribution of $X_k$.  If we had access to a ‘full batch’ of samples from each $X_t$ in this setting, then taking expectation over the gradient signal would be in perfect harmony with the problem setting.  But non-iid data with a ‘full batch’ per timestep is not realistic. So for us, the question is rearranged: "are expected gradients and the resulting deterministic differential equations the right theoretical tools for the setting?"
>
> Using expected gradients to investigate stochastic descent is not unprecedented, even in literature specifically employing differential equations.  We cite several such works, but we didn't want to rely on precedent alone, which is why we carefully ablated from expected to stochastic in Experiment 4.3.  To further assuage the reviewer, we have also run fully stochastic versions of Experiments 4.1 and 4.2, with a single sample per timestep instead of twenty.  The resonant regions of the stochastic runs for 4.1 and 4.2 align extremely well with the existing results and (more importantly) with theoretical predictions made from expected gradients.  This suggests resonance instability is well characterized by our analysis, despite integrating away the S in SGDm.  We will include these in the camera ready paper, if accepted, and are happy to share the plots here as well, if that is allowed.
>
> The closest alternative approach would be stochastic differential equations, where no expectation is taken over the gradient signal, and its continuous time limit is a transformation of the Wiener process. This is interesting, but is more complex to wield, and has much less developed theory of stability and resonance.  Considering the already excellent alignment between theory and experiment, we don't see the need to move our theory from a broad and simple foundation to one which is more narrow and complex.
>
> **Point 5.**  Your observation is correct that the worst case of resonance can be avoided by $\mu \le 0.96$, but this threshold is specific to the problem instance, and other variations of the setting (higher noise, stronger frequency peak, more frequency peaks, etc) will have different thresholds.  Appendix A.2 provides some extra discussion. It also bears mentioning that even if the worst case (exponential divergence) is easily avoided, frequency response can still be nontrivial for sub-resonant momentum values.  A practitioner may see slow convergence at a reasonable momentum value, simply because they have chosen a $\mu, \eta$ pair which resonates with their data.  Figure 6 shows this relationship. To be clear, we are not suggesting that divergence due to resonance is widespread in practice.  But there are certainly scenarios where it could occur. Many practitioners use values of $\mu$ close to 0.99, especially in early training.  Certain types of data; like audio data, machine sensors, weather data, traffic data; has power spectral densities that peak sharply around several frequencies, hence are capable of aggressively driving resonant responses.
>
> Further work is necessary to determine the severity and prevalence of resonance on natural data, and to practically characterize non-resonant values for a given setting, and we leave both as open problems.  For this paper, the scope is rigorous foundation.  Even if a practitioner can easily avoid the problem, a theoretician cannot.  To theoreticians studying SGDm under non-i.i.d. $\{X_k\}$, our result is relevant as presented, because it’s now clear that any sharp convergence result in this setting (which are currently both desirable and rare) must take resonance into account.
>
> **Point 6.**  Even on the discrete domain, the non-i.i.d. $\{X_k\}$ with SGDm induces a parametric oscillator.  You are correct that the discrete oscillator is a more direct representation of the algorithm’s iterates, and that the continuous oscillator is an approximation.  But analysis on the continuous domain is vastly easier.  Indeed, across the engineering, physics, and mathematics literature, there is far less known about the discrete time parametric oscillator.
>
> This is precisely why the continuous time dynamics lens is valuable to those studying descent algorithms.  Certain kinds of analysis are far more tractable.  Indeed, the introduction and related work sections of Muehlebach and Jordan 2021 summarize many insights and intuitions which have emerged from the continuous time domain analysis of discrete descent algorithms.  Our work is simply part of this endeavor.
>
> Any further discussion on these points is highly appreciated!

---

### Author Response · Authors · 2021-11-18
**Paper Updated**

In our responses to many of the major and minor points in each review, we state that we will make some change to the paper.  All of those changes have now been made and submitted.  The only exceptions are those changes which we explicitly propose as a change for the camera ready version, if the paper were to be accepted.

We thank all reviewers for their time and effort providing thorough feedback.  We hope all agree that the paper has been strengthened in addressing it.

We enjoyed discussing the feedback.  Should the reviewers decide to engage further, we are happy to continue!

---

### Decision · Program_Chairs · 2022-01-20

**Decision:**

Accept (Poster)

**Comment:**

This paper studies online learning using SGD with momentum for nonstationary data. For the specific setting of linear regression with Gaussian noise and oscillatory covariate shift, a linear oscillator ODE is derived that describes the dynamics of the learned parameters. This then allows analysis of convergence/divergence of learning for different settings of the learning rate and momentum. The theoretical results are validated empirically, and are shown to generalize to other settings such as those with other optimizers (Adam) or other models (neural nets). The reviewers praise the clear writing and the rigorous and systematic analysis.

3 out of 4 reviewers recommend accepting the paper. The negative reviewer does not find the main contribution interesting and significant enough for acceptance. Although I think this is a reasonable objection, it is not shared by the other 3 reviewers. Since the negative reviewer does not point out any critical flaws in the paper, I think the positive opinions should outweight the negative one in this case. I therefore recommend accepting the paper.